# The maximal and current accuracy of rigorous protein-ligand binding free energy calculations

Gregory A. Ross [1,3✉], Chao Lu[1], Guido Scarabelli[2], Steven K. Albanese[1], Evelyne Houang[1], Robert Abel[1], Edward D. Harder[1] & Lingle Wang [1✉]

Computational techniques can speed up the identification of hits and accelerate the development of candidate molecules for drug discovery. Among techniques for predicting relative binding affinities, the most consistently accurate is free energy perturbation (FEP), a class of rigorous physics-based methods. However, uncertainty remains about how accurate FEP is and can ever be. Here, we present what we believe to be the largest publicly available dataset of proteins and congeneric series of small molecules, and assess the accuracy of the leading FEP workflow. To ascertain the limit of achievable accuracy, we also survey the reproducibility of experimental relative affinity measurements. We find a wide variability in experimental accuracy and a correspondence between binding and functional assays. When careful preparation of protein and ligand structures is undertaken, FEP can achieve accuracy comparable to experimental reproducibility. Throughout, we highlight reliable protocols that can help maximize the accuracy of FEP in prospective studies.

[1] Schrödinger Inc, New York, NY, USA. [2] Schrödinger Inc, Cambridge, MA, USA. [3] Present address: Isomorphic Labs, London, UK.
✉email: gregoryaross@isomorphiclabs.com; lingle.wang@schrodinger.com

Two critical objectives in drug discovery are developing molecules that bind tightly to a target protein and weakly - or not at all - to off-target proteins. There is a growing consensus that computational methods can help identify early promising compounds and aid the otherwise slow and expensive stage of lead development. In recent years, alchemical free energy calculations[1] – a family of rigorous, physics-based methods – have emerged as the most consistently accurate method available[2–4]. The purpose of this work is to quantify how accurate these types of method has become and to ascertain how close the predictions currently are to experimental accuracy.

Relative alchemical binding free energy calculations consist of a series of simulations in which the interaction and internal energies of pairs of molecules are interpolated. Statistics collected during the course of these simulations are used to produce estimates of the difference in binding free energy between the two molecules. While robust methods for computing absolute binding free energies are emerging, these also produce relative binding free energies if the free energy difference between the *apo* and *holo* protein conformations is unknown[5,6]. The term free energy perturbation (FEP) often refers to a particular class of alchemical method[7] but has in recent years been applied to alchemical binding free energy methods more generally; henceforth, we will refer to all alchemical binding free energy methods, including thermodynamic integration, as FEP. Out of all the software programs that can perform FEP[8–15], the FEP+ computational workflow has seen the widest adoption in industry[3,16]. Although FEP+ is one program, the accuracy that it can achieve is frequently taken as an indication of what FEP methods can achieve as a whole[2].

There are numerous studies that report the successful application of FEP+ in live drug discovery projects[17–25]. Although physics-based affinity prediction methods are intrinsically computationally expensive, large clusters of graphical processing units and cloud computing enable the evaluation of relative binding free energies for potentially thousands of pairs of compounds within the typical time-constraints of medicinal chemistry deadlines. The accuracy of FEP methods have been steadily increasing due to force field improvements[26,27] and the application of enhanced sampling techniques[28,29]. Along with the increase in accuracy there has been an increase in the domain of applicability. While FEP methods are more often associated with R-group modifications[16,30], advances in methodology have meant these methods, particularly FEP+, can be applied to chemical modifications involving macrocyclization[31], scaffold-hopping[32], covalent inhibitors[33], and buried water displacement[34].

As input, FEP requires the three-dimensional structures of the protein and the putative binding geometries of the chemical series of whose absolute or relative binding free energy will be assessed. A historically difficult aspect of preparing structures has been the determination of the protonation and tautomeric states of the ligands and the protein binding residues[35]. Ambiguities in the protein structure, such as missing loops and flexible regions also pose a challenge and require careful consideration by users as to how they will be modeled. Because of these uncertainties, practitioners commonly perform a retrospective study of FEP on previously assayed compounds to test the reliability of the structural models before moving on to prospective predictions[36].

Although the main utility of FEP lies in its ability to aid live drug discovery projects, retrospective assessment studies of accuracy are, in general, vital to help identify how accurate an FEP method is and to highlight methodological aspects that require further improvement. The two largest protein-ligand benchmark data sets for FEP are the set assembled for the OPLS4 force field by Lu et al. (with a total of 512 protein-ligand pairs)[27] and the set by Hahn et al. (with 599 protein-ligand pairs)[37]; the latter of which was designed to be the community standard and has been used in the validation of the Open Force Field 2.0.0[38]. Unlike data sets such as PDBbind[39], these data sets do not contain only experimentally determined structures, and instead consist of congeneric series of ligands where all ligand binding modes and protonation states have been modeled. While these benchmarks have proved to be very useful, they currently do not cover the domain of applicability of FEP, lacking, for instance, any membrane proteins, scaffold hopping transformations, and macrocyclic transformations to name a few deficiencies. Although the OPLS4 benchmark set includes charge-charging and buried water displacing transformations, the benchmark set by Hahn et al. does not. With the benchmark set by Hahn et al., the omission of certain data sets may be by design, as only data sets that met certain quality standards were included.

The apparent accuracy of FEP is fundamentally limited by the accuracy of experimental affinity measurements. The most appropriate observables with which to compare FEP predictions against are in vitro measurements of dissociation constants ($K_d$s), inhibition constants ($K_i$s), or ligand concentrations that achieve 50% inhibition ($IC_{50}$s). While FEP predictions can be used to complement other experimental measurements that are not directly relatable to binding free energy, such as temperature shifts and percent inhibitions values, these can cloud the apparent accuracy of the predictions. By definition, the $K_i$ is the $K_d$ of an enzyme inhibitor. However, these quantities differ subtly in the way they are measured; $K_i$s are typically measured in functional enzymatic inhibition assays whereas $K_d$s come from experiments that more directly measure binding[40]. Although $IC_{50}$s are dependent on the concentrations of the protein and ligand as well as the reaction Michaelis constant, under common assay conditions and reaction mechanisms (but not all), the ratio of the $IC_{50}$s of two ligands is equivalent to the ratios of the $K_i$s[41].

There is a ladder of meaning associated with experimental accuracy that has informed prior studies on this topic. At the two lowest rungs of this ladder, one can consider accuracy as being tied to the 'repeatability' of a measurement (i.e. the same assay under the same conditions with the same equipment, usually conducted by the same team), or the reproducibility of the assay itself (i.e. the same assay set up and run by different experimenters[42,43]). After reviewing an internal repository of protein-ligand activities that had been measured multiple times, Brown et al. reported a median standard deviation between measurements of 0.3 pKi units (0.41 kcal mol)[44]. Studies have consistently found that the variance between affinity measurements made by different teams is higher compared to the variance encountered when the same team repeats the assay[45]. At the highest rung of the ladder of experimental error is the reproducibility of an affinity measurement, which is what - ideally - independent experiments could observe using different assays. To quantify the reproducibility of binding affinity measurements, Kramer et al. previously surveyed the ChEMBL database for protein-ligand complexes that had affinities measured at least twice by different groups[46]. They found that the root-mean-square difference between these independent measurements ranged from 0.56 pKi units (0.77 kcal mol$^{-1}$) to 0.69 pKi units (0.95 kcal mol$^{-1}$) depending on how the data was curated.

There are numerous factors that drive the observed differences in measured binding affinities between laboratories. These can range from concentration errors in, for instance, the reagents used in isothermal titration calorimetry experiments[47], to the difference in material of the assay containers; one study reported that a particular compound was absorbed by glass and not plastic, which artificially reduced the apparent Ki of one compound[48]. Significant differences in measured binding affinities can also occur when assays are repeated with alternate instruments or

when data are reanalyzed with different software[49]. Data fitting methods can be made more robust when using analysis methods that explicitly model uncertainties in the experiment[50,51].

While the efforts of by Brown et al. and Kramer et al. to quantify experimental error are informative, of particular interest to FEP is the reproducibility of relative binding affinities: the difference in absolute binding free energy between two molecules. One could produce such an estimate from the error from absolute affinity measurements by assuming every experimental measurement is unbiased with a Gaussian error distribution. These assumptions are significant, and, ideally, one would estimate the experimental uncertainty of relative affinities using measurements of series of compounds binding to the same protein – not single ligands – from different and independent assays. To date, no such study has been conducted despite a notable previous attempt[52].

In this study, we conduct two surveys to ascertain the accuracy of FEP. The first of these two surveys is concerned with the reproducibility of relative (as opposed to absolute) binding affinity measurements. We collect binding data from studies where the affinity of a series of compounds was measured in at least two different assays. The deviation between the relative binding affinities sets a lower bound to the error we can expect from any prediction method on large and diverse data sets. In the second survey, we collect protein-ligand structures and binding data from as many previous FEP validation studies as possible. In the assembly of this benchmark data set, we sought to include systems that cover the current domain of applicability of FEP methods. We re-assess the quality of many of the modeled structures and simulation inputs to quantify the best possible accuracy that could be achieved with the FEP+ program on these data. Presented together, we hope that both surveys provide a comprehensive picture of the maximal accuracy one could expect from FEP, the current state of accuracy of FEP, and what kinds of setup procedures are robust in prospective drug discovery settings.

## Methods

**Experimental reproducibility survey**. We set out to quantify the maximal accuracy that a relative affinity prediction method could achieve on large and diverse data sets that comprise many assay types. To do so, we searched for studies where at least two different assays were used to measure the binding affinities of the same set of compounds to the same protein. We are not concerned with the difference in the absolute binding free energies as these can include systematic errors that are ultimately irrelevant for determining which ligands bind stronger than the others. Instead, we evaluated how well pairs of assays agreed in terms of relative binding free energies and rank ordering of the compounds.

Preferably, each chemical series would be evaluated in different assays conducted by independent groups. However, as this data is

hard to come by, we collected comparative assay data where the measurements formed part of a single study. Table 1 summarizes the publicly available data that we analyzed which compared the affinity of the same compounds in at least two different assays. Expanded versions of this table, which include more assay information, are in the Supplementary Methods section of the Supplementary Information (SI), Supplementary Tables 1–3. To compliment this survey, we also collected comparative assay data from our own internal drug discovery projects. This data is summarized in Supplementary Table 4 of the SI.

We were interested to see whether binding assays, such as those that directly measure dissociation constants were, on average more or less reproducible with another binding assay compared to a functional assay, such as those that measure inhibition constants.

We only considered assays that reported the affinity of ligands using dissociation constants ($K_d$), inhibition constants ($K_i$) or the ligand concentration that achieve fifty percent inhibition ($IC_{50}$). Between any two ligands $a$ and $b$, we assumed that

$$\frac{IC_{50}^a}{IC_{50}^b} = \frac{K_i^a}{K_i^b} = \frac{K_d^a}{K_d^b}, \tag{1}$$

which is true for a wide variety, but not all, inhibitory mechanisms and assay conditions[41,53]. To aid the comparison with predictions from FEP, the pairwise error between the relative binding free energies in the different assays was also calculated. If $X$ is either an $IC_{50}$, $K_d$, or $K_i$ from the same assay, the relative binding free energy between two ligands $a$ and $b$ is given by

$$\Delta\Delta G_{ab} = -kT \ln\left(\frac{X^b}{X^a}\right), \tag{2}$$

where $k$ is the Boltzmann constant and $T$ is temperature, assumed throughout to be 300K. To compare two assays with affinity data on the same protein and set of ligands, the relative binding free energies (i.e. $\Delta\Delta G$s) were computed between all pairs of ligands within each assay, creating two sets of relative binding free energies. The root-mean-square error or mean unsigned error between these two sets of relative binding free energies provides a measure of the reproducibility of the assays. The comparison was performed for all pairs of ligands sets shown in Table 1 and Supplementary Tables 1–4.

To quantify the agreement between the rank ordering of chemical series between two different assays, we calculated the coefficient of determination ($R^2$) and the Kendall rank correlation coefficient (Kendall's $\tau$) of absolute binding free energies $\Delta G_a = kT \ln(X^a)$. When $X$ is an $IC_{50}$, these absolute binding free energies include an unknown additive constant that has no effect on rank and correlation measures.

**FEP accuracy benchmark**. We aimed to establish the most comprehensive publicly available FEP data set to date by collecting congeneric series from as many previously published FEP

**Table 1 The proteins where the same set of ligands had binding affinities measured in at least two different assays from our survey of publicly available data.**

| Comparison type | Proteins in set | No. compounds |
|---|---|---|
| Binding vs binding | SH2, Herg, Cal, CalI, Lectin, HCV polymerase, FAK, trypsin, DPPIV, bromodomains, BPTF, galectin 3 | 250 |
| Binding vs functional | Xiap, thrombin, HCV polymerase, Hsp90, FAK, DPPIV, MAPK13, AChE | 1207 |
| Functional vs functional | COT kinase, ACE, SARS-Cov2 main protease | 410 |
| | Total | 1867 |

Targets from our own drug discovery efforts are not listed but contribute to the total number of compounds in each category. The assays are categorized in terms of being 'binding' or 'functional'. In some cases, like CalI, there are multiple studies that compare different series of ligands. The number of compounds includes duplicates in the sense that the same compounds can appear in multiple assay comparisons. More details on the provenance of these data can be found in the Supplementary Methods.

studies as possible. Two aims for this benchmark data set were to include as many ligands as possible and to cover the range of targets and ligand perturbations that occur in drug discovery, such as charge changing and/or buried water displacing perturbations. By meeting these aims, we hoped that the resultant benchmark would provide a thorough test for current and future FEP methodologies. While the majority of the systems in this benchmark come from previous FEP studies, additional data sets with well-resolved protein structures and ligand binding affinities that were encountered during the assembly of this data set were also included.

For inclusion in this study, we required that an X-ray structure of the protein-ligand complex exists for at least one ligand in the congeneric series, there were no significant structural ambiguities in the protein structure, and the binding data for the congeneric series was measured as $K_d$s, $K_i$s, or $IC_{50}$s. Unlike Hahn et al.[37], we did not omit chemical series on the grounds of having too few compounds or too small a dynamic range of the experimental binding free energies. While having such requirements leads to tighter prediction error and correlation metric confidence intervals on individual chemical series, we are primarily concerned with the aggregated performance of FEP on the whole data set. Metrics such as overall RMSE or Kendall's tau can be calculated with high confidence even if the data sets they aggregate over are small or have a narrow dynamics range. We were motivated to include some of the smaller data sets because they covered specific chemical types or transformations that are not present in other larger benchmark data sets. Examples of these data sets include those focused on macrocycles[31,54], charge changing transformations[55], and buried water displacement[34].

Table 2 lists the protein-ligand data sets that were collected in this study along with the publications where they first appeared. The name given to each data set is the same as used in the Supplementary Results.

As the accuracy of FEP is dependent on the input protein and ligand structures, we endeavored to ensure all of the structures were of a consistently high quality. For the majority of systems, we reviewed the ligand binding geometries, the protonation states of ligand and binding site protein residues, the structure of the flexible regions of the protein, as well as the protein crystal structures themselves. Examples of where we changed the protein crystal structure include the CKD8 and SYK systems from Schindler et al.[56] and the JAK2 fragment set from Steinbrecher et al.[57]. The Supplementary Results details in-depth the modifications we made to the original inputs. We note that the

work by Hahn et al.[37] lists alternative crystal structures for many of the systems used in this benchmark which could be considered in future development of our benchmark set.

Where possible, modifications were made to systems that would be plausible in a prospective setting. For instance, if it was unknown which protonation or tautomeric state the ligand was in, all states were added to the FEP map so that the relative free energies could be calculated and accounted for using our pKa correction protocol[55,58]. Similarly, if the rotameric state of a ligand chemical group was unknown, either the chemical group was decoupled in the calculations to facilitate sampling or multiple rotameric states were added to the FEP map and corrected for as described in the Supplementary Methods, section 1.2. Examples of the kinds of changes and modifications we made to the systems are shown in Fig. 1.

*Simulation details.* All simulations were conducted using FEP+ within the Schrödinger software suite (versions 21-3 and 21-4) with the OPLS4 force field and the SPC water model[27]. FEP+ uses replica exchange with solute tempering[28] where exchanges between neighboring replicas are attempted every 1.2 ps. By default in FEP+, the number of lambda windows in a calculation depends on the type of perturbation; charge-changing perturbation use 24 lambda windows, scaffold hopping and macrocyclization perturbations use 16 lambda windows, and all others use 12 lambda windows. In some cases from our benchmark (see the SI) more lambda windows were used.

For alchemical transformations that changed the charge of the ligands, the total charge of the simulation box was kept constant by transmuting a Na+ or Cl- ion either to water or vice versa, depending on the charge difference and perturbation direction using the scheme previously described[55]. Neutralizing counterions and a 0.15 M concentration of NaCl were added to the simulation box for charge-changing perturbations; all other perturbation types had no counterions or salt added. Unless otherwise stated, each lambda window was simulated for 20 ns. Integration was performed using the multiple time-stepping RESPA integrator[59] and hydrogen mass repartitioning using the following time steps: 4 fs for bonded interactions, 4 fs for nonbonded interactions within the distance cutoff, and 8 fs for electrostatic interaction in reciprocal space.

For all simulations, the temperature was maintained at 300 K using the Nose-Hoover chains thermostat[60]. All complex leg simulations were run in the $\mu$VT ensemble whereby water molecules were sampled with grand canonical Monte Carlo; all

**Table 2 The data sets where the initial structures and affinities were based on.**

| Data set name | Proteins in data set | No. compounds |
|---|---|---|
| FEP+ R-group set[16] | BACE1, CDK2, JNK1, Mcl1, p38, PTP1B, thrombin, TYK2 | 199 |
| FEP+ charge-change[55] | CDK2, DLK, EGFR, EPHX2, IRAK4, ITK, JAK1, JNK1, PTP1B, TYK2 | 53 |
| OPLS stress set[27] | BACE1, CHK1, Factor Xa | 114 |
| OPLS drug discovery[27] | A, B, C, D, E | 93 |
| Water displacement[34] | BRD4(1), CHK1, Hsp90, scytalone dehydratase, TAF1(2), thrombin, urokinase | 76 |
| FEP+ Fragments[57] | T4 lysozyme, LigA, Mcl1, MUP-1, JAK-2, hsp90, p38 | 79 |
| FEP+ macrocycles[31] | BACE1, CHK1, CK2, MHT1, HSP90 | 34 |
| FEP+ scaffold-hopping[32] | BACE1, $\beta$-tryptase, CHK1, ER$\alpha$, Factor Xa, | 17 |
| Merck sets[56] | CDK8, cMet, Eg5, HIF-2$\alpha$, PFKFB3, SHP-2, SYK, TNKS2 | 264 |
| GPCRs[74,75] | A2A, OX2, P2Y1 | 98 |
| Bayer macrocycles[54] | Ftase, BRD4 | 8 |
| Janssen BACE1[36,76] | BACE1 | 74 |
| MCS docking[77] | HNE, Renin | 49 |
| Miscellaneous | CDK8[78], Galectin[10,79], BTK[80], HIV1 protease[81], FAAH[82] | 79 |
|  | Total | 1237 |

The proteins and number of ligands are shown. The citations on the data set names show the study where the initial protein and ligand structures were taken from.

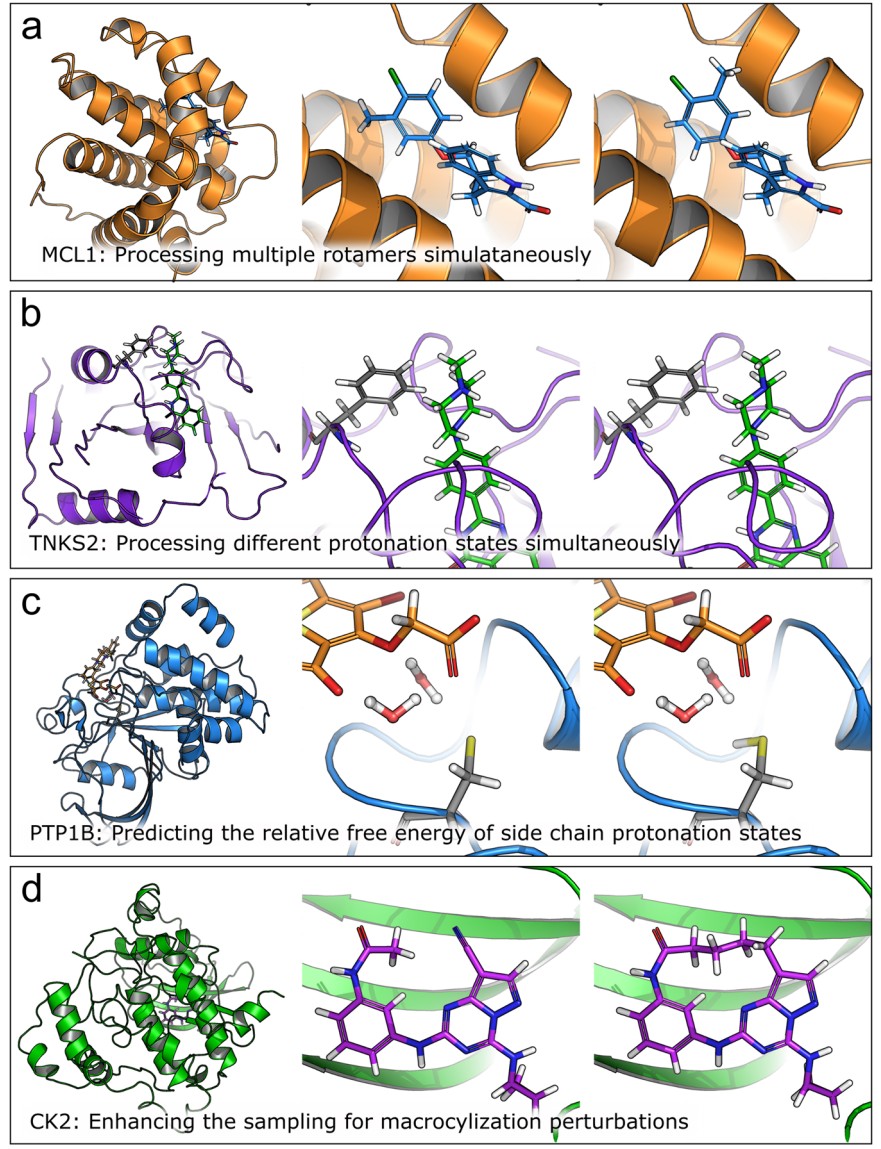

**Fig. 1 Examples from the FEP benchmark where the calculations were augmented.** Where possible, FEP was used to automatically determine the (**a**) preferred rotamer state of R-group modifications (like with MCL1), **b** protonation state of the ligand (like with TNKS2), and (**c**) the protonation state of side chain residues (like with PTP1B). **d** In some cases, (like CK2), atom mappings were altered to enhanced conformation sampling. MCL1 exemplifies cases where multiple orientations of R-groups were added to the FEP map and post-processed based on Supplementary Eq. 2 in the SI. TNKS2 exemplifies cases where multiple protonation states were post processed based on our pKa correction workflow[55]. In PTP1B, the sulfur atom is in close proximity to a number of backbone NH groups (not shown) which are likely to be the the main drivers of the predicted negatively charged protonation state.

solvent simulations were run in NPT using the Martyna-Tobias-Klein barostat[61] to maintain the pressure at 1 atm. The use of this enhanced water sampling procedure in the complex legs negated the need to assess the sensitivity of the predictions on the starting positions of water molecules[34].

When preparing proteins and ligands for FEP+, the Schrödinger protein preparation wizard was used. All crystallographic water molecules were retained and missing side chains or loops were added with Prime. Protonation state assignment was carried out with PROPKA[62] and manual inspection. A detailed description is provided in the SI for systems that required more involved preparation and analysis.

*Metrics used for analysis.* Relative FEP methods calculate the binding free energy difference ($\Delta\Delta G$) between pairs of structurally similar ligands. For a series of $N$ ligands binding to the same receptor, in principle, simulations can be performed for all of the $N \times (N-1)/2$ pairs of ligands. However, this is prohibitively expensive when $N$ is large. Instead, it is standard practice to calculate the $\Delta\Delta G$s for a small subset of the possible pairings. The set of perturbation pairs and ligands makes a graph with each perturbation pair forming an edge. Many previous studies have reported FEP accuracy using either the mean unsigned error (MUE) or root-mean-square error (RMSE) between the calculated and experimental $\Delta\Delta G$s for each edge. However, these so-called edgewise errors are dependent on the topology of the perturbation graph. As perturbation graphs are usually constructed to have edges between similar ligands, edgewise errors are correspondingly limited to quantifying FEP accuracy for similar ligands. A more robust alternative is to instead report the inferred predicted $\Delta\Delta G$ between every pair of ligands in the graph. To ensure a consistent set of $\Delta\Delta G$s between all pairs of ligands, FEP+ uses the cycle closure correction algorithm[63]; there exist other methods for doing so[64,65]. In addition to the edgewise

RMSE, we also calculated these pairwise RMSEs for every FEP map in our benchmark. For a collection of $M$ FEP graphs, we calculated the weighted average of the RMSEs using the following formula:

$$\text{RMSE} = \sqrt{\frac{1}{\sum_i^M w_i} \sum_i^M w_i \text{RMSE}_i^2}, \qquad (3)$$

where $w_i$ was the weight applied to the $i$th graph RMSE. For edgewise RMSEs, $w_i$ was set equal to the number of edges in each graph and for pairwise RMSEs $w_i$ was set equal to the number of compounds in the graph. The latter weighting was also used when computing the aggregate RMSE in the experimental survey. We note that this weighting scheme, along with the use of pairwise errors, was used in the FEP assessment by Schindler et al.[56].

Using the cycle closure correction algorithm, the absolute binding free energies ($\Delta G$s) - up to an unknown constant - for the ligands in each graph were determined. Metrics such as the $R^2$ or Kendall's $\tau$ of these $\Delta G$s do not depend on this unknown constant. Unlike correlation statistics for the predicted $\Delta \Delta G$s, correlation statistics for $\Delta G$s provide a direct measure of the rank order ability of FEP. We do not report the correlation statistics for the predicted $\Delta \Delta G$s because, as illustrated by Hahn et al.[37], these statistics are dependent on the arbitrary sign of the $\Delta \Delta G$s. A weighted average of $R^2$ or Kendall's $\tau$ of the $\Delta G$s was calculated across every graph in our data set, where the weight was equal to the number of ligands in the graph.

## Results

### Using FEP to resolve ambiguities in the structural inputs.
During the course of our FEP benchmarking exercise, we found that significant gains in accuracy could be made when different structural inputs could be all treated within the FEP workflow. The Supplementary Results in the SI, including Supplementary Tables 6–34 and Supplementary Figs. 1–21, provides extensive details of structural modifications and additional FEP calculations that were applied to each system. Figure 1 shows four examples where FEP was used to resolve ambiguities in the input structures. The top of Fig. 1 shows MCL1, where several ligands (such as ligand 35 in the Figure) had additional rotamers added to the map. Other groups have also considered alternate rotamer states for this ligand series[66]. Augmenting the map with additional rotamer states in MCL1 and applying the binding mode correction, as detailed in section 2 of the SI, reduced the pairwise RMSE from 1.41 kcal mol$^{-1}$ to 1.24 kcal mol$^{-1}$. In cases such as this, replica exchange solute tempering was not sufficient to lower the sampling barriers enough to facilitate complete rotamer sampling; extra rotamers were added to the perturbation map when poor rotamer sampling was observed in prior simulations using the automatically generated FEP+ analysis panel. Second from the top of Fig. 1 shows TNKS2, where a subset of the ligands had titratable amines (such as ligand 8a in the Figure). Adding both the protonated and deprotonated forms of these ligands to the map reduced the pairwise RMSE from 2.10 kcal mol$^{-1}$ to 1.60 kcal mol$^{-1}$.

The protonation states of side chains were also validated using the protein residue mutation functionality of FEP+. This approach is exemplified by PTP1B (second from bottom in Fig. 1). In this system, the sulfur atom of a cysteine residue sits within a bowl of backbone NH groups but it is also in close proximity to the carboxylic acid group of the ligands; this mixed electrostatic environment makes the determination of the cysteine protonation state nontrivial. In our previous FEP+ validation studies, we treated this binding site cysteine as being deprotonated (i.e. negatively charged). This decision has since been called into question[10]. Rather than choosing a particular protonation

state for CYS 215, we calculated its pKa using FEP+ in the presence of 4 representative ligands. The pKa of CYS 215 ranged from 0.92 to 1.66 across the four ligands so we continued to treat it as deprotonated in our calculations (the pairwise RMSE of the map was 0.74 kcal mol$^{-1}$).

As previously noted by Paulsen et al.[67], perturbations involving macrocyclization can benefit from reducing torsion barriers to enhance sampling and thereby more fully predict strain energy differences. We also observed such benefits in our macrocyclization calculations, such as CK2 shown at the bottom of Fig. 1. The acyclic ligand binds with an amine bond in the relatively high energy *cis* conformation, but the high torsional barrier height prevents the switching between *cis* and *trans* conformations in solvent. An atom mapping was chosen to place the amine in the alchemical region and torsion energies in the alchemical region were scaled to zero in the intermediate $\lambda$ windows. Without this particular atom mapping, the error between the acylic ligand and macrocycle was 3.4 kcal mol$^{-1}$. With the scaled torsion angle, the error reduced to 1.25 kcal mol$^{-1}$. The improvement in error comes from better capturing the relative strain energy between the acyclic and macrocyclic ligands. While the changes described here came from visual inspection and manual intervention, an automated approach for these kinds of perturbations would be preferable.

### Comparing the accuracy of FEP+ with experimental reproducibility.
To evaluate the maximal accuracy one could ever expect from relative binding free energy prediction methods, we conducted a survey on the reproducibility of experimental affinity assays using publicly available data as well as data from our own drug discovery projects. This survey focused on how well different assays agreed with regards to rank ordering and relative binding free energies (i.e. how much stronger one compound bound compared to another). Assay comparisons involved both binding assays and functional assays. Table 3 summarizes the reproducibility in terms of root-mean-square error (RMSE) and mean unsigned error (MUE) of the relative measurements, as well as with the coefficient of determination ($R^2$) and Kendall rank coefficient (Kendall's $\tau$). These metrics were aggregated by weighting each assay comparison by the number of compounds; confidence intervals were calculated by bootstrap sampling over each assay comparison.

Table 3 also shows the aggregated statistics from our large FEP benchmark after curation. The pairwise errors and correlation metrics of the experimental survey and FEP+ benchmark are directly comparable. whereas the edgewise errors only apply to the FEP predictions as they are dependent on the topology of the perturbation graphs. Given the sampling noise, uncertainty in the structural inputs, and force field error, the pairwise RMSE and MUE of FEP+ is, perhaps surprisingly, close to the experimental survey-weighted average. The weighted experimental error values are more uncertain, however, which are indicated by the larger range in the bootstrap confidence intervals. The correlation and rank statistics are higher by a statistically significant degree in the experiment survey than in the FEP+ benchmark. Figure 2 shows correlation plots from the experimental assay comparisons and FEP+ predictions against experiment that are representative of the best, average, and worst RMSE.

In both the experimental survey and FEP benchmark, we endeavored to remove measurements that were clearly below or above the assay detection limits. These kinds of data points were identifiable as vertical or horizontal lines in scatter plots. In the Covid Moonshot study[68], the largest publicly available data set in the experimental survey, this removal only modestly reduced the calculated pairwise RMSE from 0.85 kcal mol$^{-1}$ (using all 528

**Table 3 Summarizing the reproducibility of the experimental relative binding free energies and the accuracy of FEP+.**

| Accuracy metric | Experimental survey | FEP+ benchmark |
|---|---|---|
| Pairwise RMSE (kcal mol$^{-1}$) | 0.91 [0.83, 1.11] | 1.25 [1.17, 1.33] |
| Pairwise MUE (kcal mol$^{-1}$) | 0.67 [0.61, 0.83] | 0.98 [0.91, 1.05] |
| Edgewise RMSE (kcal mol$^{-1}$) | N/A | 1.17 [1.08, 1.25] |
| Edgewise MUE (kcal mol$^{-1}$) | N/A | 0.91 [0.84, 0.98] |
| $R^2$ | 0.79 [0.75, 0.82] | 0.56 [0.51, 0.60] |
| Kendall $\tau$ | 0.71 [0.65, 0.74] | 0.51 [0.48, 0.55] |

The value of every metric, such as RMSE or $R^2$, is a weighted average. For the pairwise, $R^2$, and Kendall $\tau$ metrics, the weighting is equal to the number of compounds in the assay (in the experimental survey) or FEP graph. For the edgewise errors, the weighting is equal to the number of edges in each FEP graph. Square brackets encompass 95% confidence intervals that have been calculated by bootstrap sampling over the pairs of experimental series or FEP+ graphs. As the edgewise error is dependent on the topology of an FEP+ graph, there is no equivalent metric in the experimental survey.

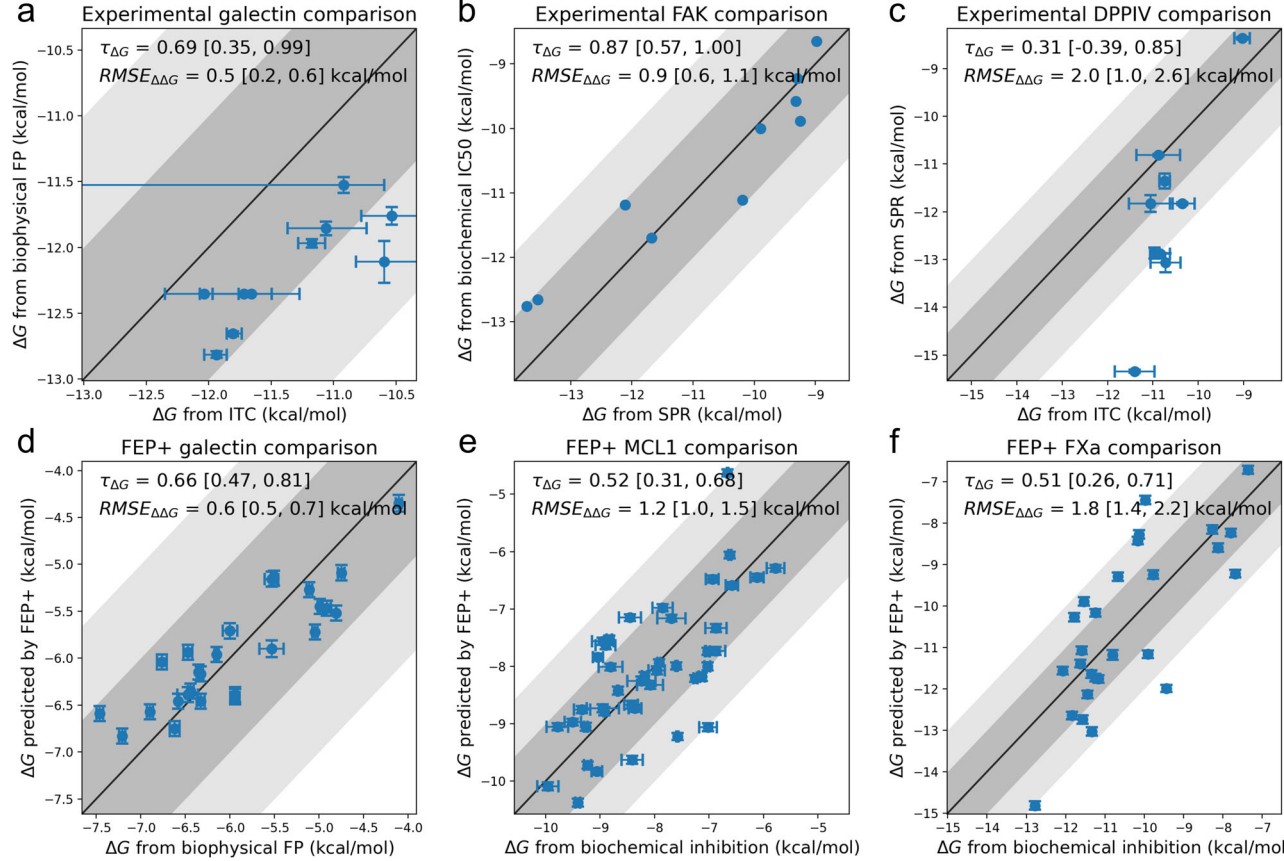

**Fig. 2 Scatter plots showing the range of agreement of ΔGs between different experimental assays and FEP+ predictions.** Panels **a**–**c** compare measured affinities between two different experiments and panels **d**–**f** compare FEP+ predictions (y-axis) against experimental measurements (x-axis). Panels **a** and **d** show examples where the pairwise RMSE of relative binding free energies was much better than average, panels **b** and **e** show examples where the RMSE was close to the average, and panels **c** and **f** show examples where the RMSE was worse than average. The top left of each plot shows Kendall $\tau$ and pairwise RMSE for each data set. Points in the dark gray area are measurements or predictions that are within 1 kcal mol$^{-1}$ of each other, and points in the light gray area agree within 2 kcal mol$^{-1}$. **a** Shows that isothermal titration calorimetry (ITC) and fluorescence polarization (FP) binding free energy measurements of galectin ligands are offset by ~1 kcal mol$^{-1}$ - this offset is irrelevant for rank ordering and does not affect the correlation or the pairwise RMSE metric. The offset of the FEP+ ΔG predictions was determined by ensuring the mean of the ΔGs was equal to the mean of experimental ΔGs on the x-axis. Where the data was available, we included the reported standard error of the experimental measurements; the standard errors from FEP+ as calculated with the cycle-closure algorithm are also indicated on the bottom row of plots.

pairs of measurements) to 0.79 kcal mol$^{-1}$ (using 324 pairs of measurements). Although measurement accuracy is generally lower close to, but not beyond, assay detection limits, we did not try to correct for these more subtle cases as we expect these will affect both the experimental survey and FEP benchmark similarly.

FEP accuracy varies depending on the method used for the calculation. To quantify this variability and the degree to which this is affected by the size and heterogeneity of the data set, we re-

analysed the predictions from FEP+ predictions from 2015 up to 2021, which encompasses sampling and force field improvements. The results, in section 2.3 of the Supplementary Results, indicate that larger, more diverse data sets are more able to discriminate between the accuracy of different approaches.

*The distribution of errors.* Figure 3 shows boxplots of the RMSEs from each assay comparison in the experimental survey and each FEP graph from the benchmark. Notably, the experimental survey

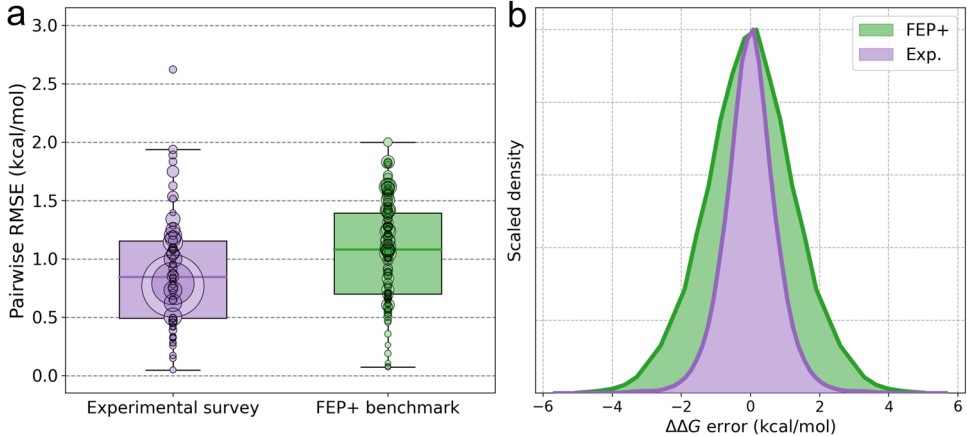

**Fig. 3 Summarizing the error distributions in the experimental survey and FEP benchmark. a** Boxplots comparing the root-mean-square error (RMSE) between relative binding free energies from different experimental assays (purple) and the FEP+ predictions against experimental data (green). The top and bottom of the boxes represent the 25th and 75th percentiles and the dark line represents the median. The whiskers extend to a maximum of 1.5 times the interquartile range. Circles are the RMSEs from comparing two experimental assays (green) or the RMSEs of a FEP+ perturbation graph (purple). The size of each circle is proportional to the number of ligands in the series in either an assay comparison or perturbation graph. The two largest data points in the experimental survey are from the COVID moonshot project[68] and project A from Supplementary Table 4. The median RMSE in the experimental survey is 0.85 kcal mol$^{-1}$ and the median in the FEP+ benchmark is 1.08 kcal mol$^{-1}$. **b** All pairwise relative binding free energy differences from the experimental survey and all pairwise FEP+ errors. The histograms were symmetrized about the x=0 line in the sense that all $N \times (N - 1)$ pairs of compounds were used. The error distributions are bell-shaped and can be approximated by t-distributions.

contains one assay comparison that has an RMSE >2.5 kcal mol$^{-1}$, which is larger than any of the errors encountered in the FEP benchmark. This data point is a comparison between surface plasmon resonance (SPR) and mass spectrometry on a series of carbonic anhydrase I ligands. Clearly, some assays can differ widely in their measured relative affinities, and so, one should expect some FEP graphs to have a large apparent error based on experimental error alone. While some graphs have RMSEs near 2 kcal mol$^{-1}$ in our FEP benchmark (see Fig. 2), none approach 2.5 kcal mol$^{-1}$ as in the experimental survey. The boxplots show that, while Table 3 states that the mean pairwise error of FEP+ across the whole benchmark data set is 1.26 kcal mol$^{-1}$, the error of FEP+ on individual graphs may be lower or higher than this value.

The right panel of Fig. 3 shows the histograms of all pairwise (not edgewise) errors from both the experimental survey and the FEP benchmark. In all 13,732 FEP+ relative binding free energy predictions, 57.5% were within 1 kcal mol$^{-1}$ of the experimental value and 12.9% differed by more than 2 kcal mol$^{-1}$, which are close – but not equal to – the percentages one would expect from a Gaussian distribution that has a standard deviation equal to the pairwise RMSE of FEP+. Out of 314,535 relative binding free energies in the experimental survey, 83.0% were within 1 kcal mol$^{-1}$ of each other and 2.1% differed by more than 2 kcal mol$^{-1}$; there are almost twice as many errors above 2 kcal mol$^{-1}$ than would be expected from a Gaussian distribution. The experimental error distribution therefore has a "fatter" tail than a Gaussian distribution that is better modeled using a t-distribution. The maximum likelihood estimate of the *degrees of freedom* parameter of the t-distribution for the experimental data was 6.0, which was lower than the estimated value of 25.7 for FEP+ errors. A smaller degrees of freedom value implies fatter tails, meaning that, although the experimental error distribution is tighter than the FEP+ error distribution, there was a higher propensity for significant, non-Gaussian differences in the experimental survey.

*Validation of the experimental error estimates.* It is worth quantifying how our aggregated experimental RMSE compares to previous estimates. After trawling the ChEMBL database for binding affinities that had been measured by at least two different groups, Kramer et al. arrived at two estimates for experimental error[46]. When all pairs of measurements were included in their estimate, they calculated the reproducibility error of absolute binding free energies to be 0.69 pKi (0.95 kcal mol$^{-1}$). Their second error estimate of 0.56 pKi (0.77 kcal mol$^{-1}$) was arrived at by discarding all differences that were >2.5 pKi units. If one assumes experimental error to be unbiased (i.e. there is no offset between two sets of $\Delta Gs$) and Gaussian distributed, then the estimates of Kramer et al. imply pairwise RMSEs of 1.34 kcal mol$^{-1}$ and 1.09 kcal mol$^{-1}$, which are arrived at by multiplying their estimates by the square root of 2. The second of these estimates is in close agreement with our estimated value in Table 3 but the first is above our upper 95% confidence limit. Nevertheless, these assumptions used to derive pairwise RMSEs from Kramer's estimates should be viewed cautiously as we have found that in some of our assay comparisons, such as in top-left scatter plot in Fig. 2, one set of experimental $\Delta Gs$ can be offset from the other. Also, as described above, we have found our pairwise error distribution is better modeled as t-distribution rather than a Gaussian. Indeed, Kramer et al. originally noted that the absolute $\Delta G$ error distribution from the ChEMBL set was poorly approximated by a single Gaussian distribution.

Our publicly available experimental survey contains comparative assay studies from a wide range of academic and industrial laboratories, all of which may have different levels of quality control. On the other hand, data from our own drug discovery projects should be more consistent in terms of quality control. To see whether this is reflected in the experimental error estimates, we can split the data sets and calculate the experimental error separately. Using only the data from our internal projects, we calculate the weighted RMSE to be 0.88 [0.80, 1.13] kcal mol$^{-1}$, compared to 0.96 [0.83, 1.24] kcal mol$^{-1}$ from the publicly available data. As the bootstrap confidence intervals overlap substantially, we cannot distinguish between the quality of the data sources. The broad consistency of these estimates suggest that experimental error values shown in Table 3 are generally representative.

The primary interest of our experimental survey is to quantify the reproducibility of experimental binding affinities, which we have approached by quantifying the difference of (relative) binding affinity measurements from different experiments. Nevertheless, it is of interest to place the experimental RMSEs in context of the uncertainty that arises from multiple repeats of the same experiment. In Supplementary Table 5 of the SI, we collect the reported standard deviations from repeats from a total of 350 compounds spread over 15 experimental assays. The overall root-mean-square of these reported standard deviations is 0.23 [0.18, 0.33] kcal mol$^{-1}$, where the square brackets denote 95% confidence intervals that have been calculated with bootstrap sampling over the different assays. The uncertainty from the repeats of a single experiment contributes to RMSE when comparing different experiments. Assuming the repeatability error we have calculated is Gaussian distributed and applies to all experiments, the RMSE from taking the difference from two measurements is approximately 0.33 kcal mol$^{-1}$ (from the square root of 2 times 0.23 kcal mol$^{-1}$). This value is roughly a third of the 0.91 kcal mol$^{-1}$ reproducibility RMSE from Table 2, which implies that two thirds of the reproducibility RMSE comes from intrinsic differences from the different experiments.

*The variability between different assay types.* Our experimental survey also permitted an assessment on the agreement between binding assays, such as surface plasmon resonance (SPR), and functional assays, such as enzymatic activity assays. As FEP predicts relative binding free energies, one could expect a better apparent accuracy with binding assays than with functional assays. Previously, Schindler et al. found that with one protein and chemical series, the predictions from FEP+ were in closer agreement to the measurements from SPR than from a functional assay[56].

Table 4 compares the experimental reproducibility between binding and functional assays and combines the data from our experimental error survey (Supplementary Tables 1-3) and from our internal program analysis (Supplementary Table 4). When ignoring the confidence intervals, the pairwise RMSE between binding assay derived $\Delta\Delta G$s and functional assay derived $\Delta\Delta G$s appears lower than the pairwise RMSE between two binding assays. This is a result of project A from Supplementary Table 4 dominating the weighted mean because of its large number of compounds. The confidence intervals, as with all others in this study, have been calculated by bootstrap sampling over each assay comparison and minimize the effect of any one comparison. Clearly, there is substantial overlap between the confidence intervals between the 'binding vs binding' and 'binding vs functional' comparisons. Thus, these data support the hypothesis – summarized in Equation (1)– that $\Delta\Delta G$s from binding assays are consistent with $\Delta\Delta G$s from functional assays when considering a diverse range of experiments and targets. With regards to what data is best for validating relative binding free energies predictions from FEP, these results show that, in the main, functional measurements of affinity are as appropriate as binding measurements. Having more comparative assay data may change

these conclusions, and these results do not preclude the existence of large differences between binding and functional assays that occur on a case-by-case basis, such as with project B in Supplementary Table 4.

Table 4 suggests that functional assays are more consistent with each other than they are with binding assays. However, there are far fewer comparisons in the "functional vs functional" category, which makes these weighted means and confidence intervals less meaningful than the others.

## Discussion

We have assembled what is to our knowledge the largest benchmark dataset for free energy perturbation (FEP) calculations of relative binding free energies to date. As prediction accuracy is only meaningful in the context of experimental accuracy, we conducted a survey of experimental reproducibility alongside our FEP benchmark. While other studies have looked at the experimental differences in absolute binding free energies[44,46], our survey focused on reproducing rank ordering and relative binding free energy measurements. Our survey used sets of ligands that had binding affinities determined in at least two different assays.

We found that the accuracy of FEP+ was close to experimental accuracy in terms of relative binding free energies while experimental rank ordering ability was found to be markedly higher than FEP+. As extensively detailed in the Supplementary Information and summarized in Fig. 1, our FEP+ results were obtained by preparing and studying the systems as thoroughly as time constraints allowed. These involved augmenting the FEP graphs with additional protomeric and tautomeric states, as well as with adding additional rotamer states of R-groups, or choosing perturbation mappings that enhanced sampling. Other changes involved improving the binding modes or protein structure. These kinds of treatments, while sometimes straightforward in retrospect, can be difficult to tease out in drug discovery programs with time constraints, where predictions must come out in a timely fashion for synthesis and assaying. However, FEP is an "all-in-one" method that can predict protonation states, ligand poses, and binding affinities, so a strategy that is appropriate for drug discovery is to enumerate FEP maps with all likely states and use post-processing, as we have here, to correct for the different protonation and rotamer states.

For our FEP benchmark, we primarily collected data sets from previous publications on FEP. This potentially biases our estimates of FEP error as prior publications are more likely to contain systems where a particular FEP method appeared at least satisfactory in terms of accuracy. It should also be noted that our estimate for the error of FEP+ with respect to experiment was estimated retrospectively and that reported errors are typically higher in prospective applications[56]. These biases may partly reflect the results in Fig. 3, where the largest pairwise error was found in a comparison of different experimental assays and not FEP graphs. Our experimental survey clearly demonstrates that some experimental affinity measurements can differ substantially

**Table 4 Comparing the agreement between binding and functional assay measurements of relative binding free energies.**

|  | No. comparisons | Pairwise RMSE (kcal mol$^{-1}$) | $R^2$ | Kendall $\tau$ |
|---|---|---|---|---|
| Binding vs binding | 26 | 1.10 [0.85, 1.34] | 0.76 [0.65, 0.84] | 0.69 [0.61, 0.76] |
| Binding vs functional | 30 | 0.93 [0.82, 1.21] | 0.81 [0.73, 0.83] | 0.76 [0.62, 0.75] |
| Functional vs functional | 6 | 0.75 [0.53, 0.79] | 0.78 [0.75, 0.91] | 0.70 [0.58, 0.81] |

As each assay type differs in what is measured, in the sense that binding may not always result in inhibition, one may expect a larger disagreement between the two types than within the types. The confidence intervals, calculated by bootstrap sampling over the different assay comparisons, show that the differences we have obtained are not statistically significant. The "No. comparisons" column shows how many assays were compared to estimate the reproducibility and bootstrapped over to estimate the confidence intervals.

from others. FEP will have an apparently high error if compared against irreproducible experimental affinity measurements, and these cases will surely be encountered by increasing the amount of data in the FEP benchmark. It should also be noted that given the heterogeneity of the assay quality used both in the experimental survey and FEP benchmark, our estimates of the maximal and current accuracy primarily apply to this same regime of assay quality. While our benchmark is the largest to date, we will continue our efforts to improve its size and representation of drug discovery targets.

FEP is most useful when it can be applied to the kinds of designs that are made by medicinal chemists, which is why we have included a wide variety of transformation types in our benchmark, such as those involving fragments, scaffold-hopping, charge-change, macrocyclization, and water displacement. The range of systems and transformations contained in our benchmark covers a wider domain of applicability than previous benchmark sets, namely the OPLS4 set[27] and the set by Hahn et al.[37]. Our FEP benchmark set also contains over twice as many ligands as either of these data sets. With these considerations in mind, we hope our benchmark dataset provides the most comprehensive test of an FEP method to date. It is also our hope that FEP benchmark datasets will continue to grow in size, ligand diversity, and target coverage. The benchmark data set we present here is a step forward in this respect, although there remain areas for improvement. The number of membrane proteins in this benchmark set could be increased in future iterations to better reflect the distribution of drug targets, but we note that the aforementioned prior benchmark sets did not contain any membrane proteins. Other types of ligand transformations, such as those involving transition-metal complexes which are actively being developed were omitted from this dataset[69] and should be included in future efforts. Although we reviewed the quality of many of the protein and ligand structures in this work, this aspect of the data is an evolutionary process and the structures should remain under review in future versions of this benchmark.

While the focus of this work has been on the analysis of binding free energies for protein-ligand complexes, it is important to acknowledge that these free energy calculations have a wide range of other applications, such as estimating covalent reaction kinetics[33], small molecule solubilities[70], protein and ligand pKas[58,71], and the stability of protein mutations[13,72,73]. In each application, we believe it is important to frame reported accuracies in the context of experimental uncertainty. To help encourage more complete validations of free energy methods in future, as part of this manuscript, we are releasing the publicly available protein and ligand structures that were used in this benchmark as well as the data from our experimental accuracy survey.

As our experimental reproducibility survey indicated that experimental RMSE is on average ~1 kcal mol$^{-1}$, it would be an extraordinary challenge to ever have an FEP method that achieves an error truly statistically indistinguishable from the experimental error on a large and diverse data set, such as we have produced here. However, given the continual development of new methods and protocols, it is an open question as to how close predictions could ever get to the limits set by the inconsistent quality of the target experimental data. In the meantime, given our experiences in assembling and curating this benchmark, the greatest gains in accuracy that present day users can achieve will come from following the best practices that we have attempted to further codify here.

## Data availability
The assembled FEP benchmark data set and the publicly available experimental survey data are freely available on the Github repository github.com/schrodinger/public_

binding_free_energy_benchmark. The results in this manuscript use v1.0 of the repository. The Supplementary Information contains many details relating to the experimental error survey and FEP benchmark. The experimental survey data in the Supplementary Methods section includes tables for each assay comparison time that state the source, protein, and assay types for the publicly available data. Summary tables of Schrödinger's in-house drug discovery experimental error data are also summarized in Supplementary Table 4, although the raw data itself has not been made available on the Github repository. The Supplementary Results of the FEP benchmark contains discussions and validation results on all systems and chemical series that were modified relative to the original publication. The SI contains figures that summarize our work on each system, tables that summarize the accuracy of FEP+ on each series, and tables that contain additional analyses, such as ligand and protein pKa calculations.

## Code availability
The code to perform the error analysis as described herein is freely available on the Github repository github.com/schrodinger/public_binding_free_energy_benchmark. The results in this manuscript use v1.0 of the repository.

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

## Acknowledgements

We are very grateful to Francesca Deflorian, Gary Tresadern, Mark Seierstad, Thomas Steinbrecher, Haoyu Yu, Wei Chen, Mikolai Fajer, Francesca Moraca, and Antonjia Kuzmanic for their input and advice on the FEP benchmark data set.

## Author contributions

G.A.R. and L.W. conceived the concept with input from R.A. and E.D.H.; G.A.R. assembled and simulated the FEP benchmark data set with assistance from C.L and G.S. FEP dataset curation was performed by G.A.R with guidance from L.W. and E.D.H.; G.A.R assembled the publicly available experimental reproducibility data; S.K.A and E.H. provided the experimental reproducibility data from in-house drug discovery programs. G.A.R. performed all the analyses. G.A.R. wrote the manuscript with help from L.W. All authors discussed the results and commented on the manuscript.

## Competing interests

The authors declare no competing interests.
