## [Peer Review File · Communications Chemistry]

Reviewers' comments:

Reviewer #1 (Remarks to the Author):

Overall, this is a very interesting and potentially very valuable study from Ross et al. at Schrodinger. Its two most important contributions are (a) an assessment of the reliability of experimental affinity measurements across multiple tests or assays, and (b) a large set of benchmarks and benchmark systems for binding free energy calculations. Both of these, and especially the benchmark systems, will be extremely valuable to the community (as has the much smaller set of benchmarks available in Schrodinger's 2015 JACS paper, what's commonly called "the JACS set"). However, some additional work is needed to improve the data and provenance here before this material is quite ready for publication.

Major comments:

- In terms of selecting systems for benchmarking FEP, or in categorizing systems on which FEP is benchmarked, the authors really ought to make contact with the "Best Practices" work by Hahn et al., (LiveCoMS, <https://livecomsjournal.org/index.php/livecoms/article/view/v4i1e1497>) which is intended to provide community standards for FEP benchmarks. The paper is CITED here (with missing details) but there is no mention of it or discussion of it, which seems strange given that it seems to represent the current community consensus on the topic. Even if the authors disagree, they should discuss it and explain where they disagree/why. Additionally, that study (e.g. Table 1) looks at quality metrics concerning choice of protein structure, gives examples concerning common problems with protein structures, and gives a checklist for assessing whether problems with structures would be likely to affect results. These issues should certainly be considered and reported for the targets considered here.
- Relatedly, careful reporting of what PDB codes are used for studies is essential, but these are not always listed; for example, the SI section on Jnk1 does not report what PDB code was used. Some prior work on Jnk1 had used code 2GMX, which is a structure with considerable problems. What was used here? Somewhat relatedly, the 5HNB structure is used for the study here, but this was called out by the benchmarking study (noted above) as being of lower quality as some other structures such as 5XS2, 4CRL and others. Given this, the authors had to build in unresolved loops as they note -- "The latter two missing loops may affect the accuracy of FEP..." Why were the better structures recommended by the earlier work not used? (This may hold true across other targets as well; CDK8 was only the first example I checked against Table 1 in the benchmark study; I suggest the authors carefully go through all systems in that table).
- The same benchmarking study recommends minimum standards for dataset size and dynamic range for benchmarking FEP; again, those standards seem to be ignored here (e.g. for Ftase and BACE in Table 2 have particularly low counts; I haven't checked dynamic range yet but the authors likely should check)
- The same study provided a python package to report standardized analysis statistics for exactly this type of benchmark; again, the authors should either use the package, report the same statistics (and provide their own implementation), or clearly justify why they are not. For future work, it's essential that benchmarking studies like this be able to be compared, and if each study implements its own analysis and doesn't release the code, it becomes extremely difficult to compare, hence the need for standardization.
- Because benchmark results are so sensitive to setup details, it's essential that the inputs for the reported calculations be provided in a useful and reusable way, with clear documentation, so that

others can build on this work. While some, or much of, the data seems to be provided it is not completely clear how to use it. For example, the provided ZIP file contains a subdirectory on `fep_benchmark_data`, which has no documentation (e.g. README.md files) or metadata. Guessing, it seems safe to assume that the subdirectories there are organized by protein-ligand set as laid out in the SI, but the names don't really correspond, e.g. there's a "jacs_set" directory which contains ... what? And what's "misc"? Or "waterset"? This material should be compiled in a way that it clearly corresponds to what is laid out in the paper AND has metadata explaining provenance, otherwise it is both (a) irreproducible, and (b) very hard for anyone else to reuse for anything. I would have similar concerns with the experimental survey data, though I've not looked at it for as long. (However, I note that much of the data there is .csv files without column headers which have names like "aaron2010_spr_itc.csv"; presumably these need (a) column headers, and (b) metadata properly citing the papers, explaining details of where the data came from, and detailing how it is laid out. The authors should consult the FAIR data standards (e.g. https://en.wikipedia.org/wiki/FAIR_data) and at least attempt to adhere to SOME of these standards; at present, I am not convinced even a human expert could adequately interpret the provided data package. (As a concrete example of this concern, if I wanted to reproduce Figure 3 as the first step of extending this analysis, exactly what data should I analyze?)

- The authors should consider what they would want to happen to this dataset of curation/correction is needed, e.g. if someone finds that one of the ligands used in the study is not the one which was actually studied experimentally. Do they wish to submit an erratum to correct the structure? (Or, likewise, if there is a typo in one of the experimental values given in the SI data package). One approach to this problem would be to provide the data in a GitHub repository under version control rather than in files buried in the SI of a paper (where they can only be changed/corrected by submitting an erratum). This would allow the data used in the paper to correspond to one specific version of the repository, but still allow room for further updates and corrections.

- There seems to be no discussion, either in the paper or in the SI, of what is provided in the supporting zip file, so a reader not looking for it might not be even aware it is present.

While some of the above remarks concern dataset organization and quality and might be seen as a relatively minor issue, this paper presents the benchmark data set as a core deliverable/outcome, e.g. "We have assembled ... the largest benchmark data set to date...". If this is to serve as a benchmark data set, it should be better organized, and have appropriate metadata and provenance information, and the authors should better document the choices made in selecting their specific protein structures, etc., especially in view of the prior work on best practices for benchmarking these methods. There are critical choices which should also be documented, e.g. how were any missing residues or loops handled, were internal waters retained or removed, what protonation states (for proteins and ligands) were chosen and why, how were ligands posed, etc. Some of these issues are discussed somewhat in the SI, but not completely, e.g. the Cmet section says that 8 ligands were re-docked, but doesn't list WHICH 8 ligands were re-docked. The Syk section notes that "several of the ligands were redocked", but not which. I understand that a great deal of work went in to this, but more thorough explanation/provenance information is needed if this is to be of general value as a benchmark set.

Indeed, similar issues have affected reuse of data from the 2015 "JACS set" paper, also from Schrodinger. Structures were deposited, and have been broadly reused with some benefit to the community, but also have left many questions unanswered. Some systems retained buried waters,

others removed them, some built in missing loops, others didn't, and many choices were left undocumented with answers spreading only through the grapevine by way of human experts who have spent a great deal of time with the data. In the present study, the authors have an opportunity to recognize that (as they likely hope) this data will be broadly reused by the community and come up with a better plan for documenting the protocols used in preparing it and addressing any questions and concerns that may arise. Again, this may argue towards making the data available in a GitHub repository where subsequent questions can be answered (e.g. if someone has a specific question about a particular choice the authors forgot to document, it can be answered once-for-all in the relevant GitHub repository) and corrections made if necessary. I don't think the authors have to be compelled to do this, but certainly something better than the current data (lack of) structure is warranted.

Minor remarks:

- The references are very heavy on Schrodinger work; perhaps good to add more non-Schrodinger citations
- p2 concerning domain of applicability in benchmark sets: What is the evidence for this assertion (that the domain of applicability is not covered adequately in present sets)? The paper also never seems to revisit the idea of domain of applicability later, either, so it is unclear that the present study does a better job covering the full domain of applicability. (The Discussion notes some reasons why the data collected here might be biased towards particularly well-performing cases, which would suggest that this set does not yet probe the edges of the domain of applicability.)
- There are missing references, e.g. p9 bottom refers to "Schindler et al." but no reference is given and a search for the name "Schindler" only turns up reference 32, which is not the intended paper. (The authors may have meant the recent large scale benchmark study led by Schindler, which certainly should be cited here.)
- page 2 near bottom, notes studies across within-team vs across-team assays. I believe there is also work which separates out the effects of "fresh solutions" vs "reused solutions", e.g. a same-team experiment could use a fresh prep of all the solutions or reuse the solutions. I know this had been a major point of some of John Chodera's work on ITC, e.g. that teams don't adequately account for concentration uncertainty when reporting experimental uncertainties. It would be nice to see some discussion of the potential sources of uncertainty here.
- Relatedly, probably the prior experimental ITC cross-comparison (looking at ITC done on the same systems across labs, and the reported free energy, enthalpy and entropy) should be discussed here. I don't know the reference, but I know it's discussed in some of Chodera's work on ITC.

Reviewer #2 (Remarks to the Author):

Ross et al. present an investigation into accuracy of experimental binding affinity determination and compare this to an evaluation of predicted binding affinities using free energy calculations. They collect a large dataset for free energy calculations from the literature and report performance of the FEP+ method on this based on optimized input structures.

The investigation into experimental accuracy for binding/functional assays is a good extension of previous work by Brown et al. and Kramer et al. and will be very useful for the field. The compilation of

the dataset for evaluating free energy calculations fits well with other recent efforts in the community and will be of interest to many scientists in this area. Similarly the case by studies provide interesting learnings on best practices. I recommend the article for publication with minor suggestions for modifications.

Comments:

- Is there a plan to include this new dataset/parts of it into the Protein-Ligand benchmark at <https://github.com/openforcefield/protein-ligand-benchmark>? There are already efforts ongoing to rework some of the datasets that were included here.
- Could the authors add the error bars for the FEP predictions and potentially if reported for the experimental data in Figure 2?
- Please add the confidence intervals to the statistics in Figure 2.
- Reference 32 is incomplete.
- Reference to Koehler et al. missing in SI page 5.
- The individual results for all datasets are scattered across the SI. Could the authors collect these in a single table or provide a single table in addition to the smaller tables?
- Typo in title of Table S10

Reviewer #3 (Remarks to the Author):

Summary and recommendation

The work collects a large number of binding affinity data measured in experiments and evaluates the uncertainty in measured free energy differences from cross-experiment comparison. Further, another large set of binding affinities is estimated by means of free energy calculations. Comparing calculation results to the experimental measurement showcases that calculation reaches nearly the experimental error.

Overall, this is a very good work: well structured, executed and written. I have three main points that in my opinion would still need to be addressed:

- Given that the work already provides comparisons of experiment_vs_experiment and computation_vs_experiment, it is natural to be interested in the error between two computed results, i.e. computation_vs_computation. The authors have now calculated free energy differences following their best practice in structure and simulation preparation. However, most (maybe all?) of those values have already been evaluated previously. Would it be reasonable to consider the previous calculations as independent experiments, where researchers might have followed a slightly different protocol and assess what is the level of reproducibility for calculated $\Delta\Delta G$ s?
- It would be interesting to also provide the average uncertainty reported with the experimental measurement itself (understandably, not every published measurement has an associated uncertainty, but certainly a fraction of them do report it). It would be great to know how much of the experiment_vs_experiment error can be accounted for by the uncertainties reported by the experiments themselves. And similarly, FEP+ also reports an error associated to each individual free energy calculation. How does this uncertainty compare to the difference observed from computation_vs_computation?
- The authors did a very thorough job in going through a large number of previously assembled FEP datasets identifying the outliers and fixing or discarding them. Such a retrospective analysis of the outlying calculation results naturally lowers the computation_vs_experiment RMSE, but also makes it not fully representative of the deviation that could be expected in a prospective prediction. I think it would be important to emphasize this point, as this RMSE will not necessarily directly transfer to the prospective drug design projects. In addition, would such a rigorous outlier analysis also reduce the experiment_vs_experiment RMSE? Currently, the authors rather excluded very similar experimental setups (as described in the SI page 3), so is the experiment_vs_experiment RMSE somewhat skewed towards the larger values?

Below I also collected several minor questions and comments. After addressing these questions and comments I would recommend the article for publication in Communications Chemistry.

Introduction

On page 2 it is mentioned that a number of programs can perform FEP. Among them Amber based calculations are referenced. To my knowledge Amber uses TI rather than FEP.

Methods

For self-consistency of the manuscript it would be useful to have a bit more elaborate description of the simulation methods. In particular, charge changing modifications are known to be challenging. Which strategy did the authors use: was the overall neutral charge of the box conserved during the simulations or was a post-hoc correction applied?

Using FEP to resolve ambiguities in the structural inputs

To include multiple rotamers into a perturbation map (and later apply equation 4) the necessary condition is for the rotamers not to interconvert between the states during the alchemical calculation. How did the authors ensure that: or was it just an observation after the simulation? Was the REST not sufficient to lower the transition barriers for the rotamer interconversion?

For the PTP1B case, what would be the interpretation for such a low cysteine pKa in proximity to a carboxyl group on the ligand? Is it not unfavourable to have two partial negative charges close to one another when cysteine is deprotonated?

For the CK2 cyclization example, from a practical point of view, how one should identify in advance which barriers will not be crossed and thus need a special treatment? How does one in practice change atom mapping in FEP+?

Figure 2

It would be good to specify that ΔG was offset with the experimental mean of *the same experiment* to which FEP+ is compared to. Otherwise, it may be ambiguous which experimental set to choose, e.g. for the galectin case taking ITC mean would offset the values with respect to the biophysical FP experiment.

Figure 3

What do the boxes and whiskers represent? Are the whiskers 95% CI for the whole pairwise RMSEs (seems somewhat confusing as all the visualized points fall within the whisker range for FEP+)?

For the histograms Figure 3 (right panel) would it make sense to note whether the pairwise $\Delta\Delta G$ in both directions were considered? In the earlier RMSE calculations, as I understand $N \times (N - 1) / 2$ pairs were considered, but for this figure the signed errors were explored, so probably all $N \times (N - 1)$ were used?.

Also, the text on page 8 explains: "the percentages one would expect from a Gaussian distribution with the same pairwise RMSE". Is it rather meant that a Gaussian with the mean and variance of the $\Delta\Delta G$ error distribution from Figure 3 was considered?

Appendix

The first sentence in A.1 is confusing. It states that equation (4) is for R-groups with rotational symmetry, but equation (4) is general and can be applied to any conformations. Probably it is meant that a special case of R-groups with rotational symmetry is discussed in A.1, but then some clearer wording would help.

Text

Page 2, first paragraph: "if the energy difference" should be "free energy difference"

Page 3, third paragraph: "those that measures" should be "measure"

Page 5, second paragraph: "all others uses 12" should be "use"

Page 6, last paragraph: "Pauslon et al" should be "Paulsen et al"

Page 7, last paragraph: "that has an RMSEs" could be "that has an RMSE"

Page 8, first paragraph: "based experimental error" could be "based on experimental error"

Page 8, last paragraph: "a t-distributions" could be "a t-distribution"

Page 9, first paragraph: "imply a pairwise RMSEs" could be "imply pairwise RMSEs"

Page 9, first paragraph: "top-right scatter plot" should it say "top-left"?

Page 10, first paragraph: "lower than the pairwise RMSE between two functional assays" should it say "binding assays"?

Page 10, table caption: "differences we have obtained is not" should be "are not"

Page 10, last paragraph: "as the data as prior publications" probably "as the data" is not needed here

Page 12, first paragraph: "absolute binding free protocol" should be "absolute binding free energy protocol"

Reviewers' comments:

Reviewer #1 (Remarks to the Author):

Overall, this is a very interesting and potentially very valuable study from Ross et al. at Schrodinger. Its two most important contributions are (a) an assessment of the reliability of experimental affinity measurements across multiple tests or assays, and (b) a large set of benchmarks and benchmark systems for binding free energy calculations. Both of these, and especially the benchmark systems, will be extremely valuable to the community (as has the much smaller set of benchmarks available in Schrodinger's 2015 JACS paper, what's commonly called "the JACS set"). However, some additional work is needed to improve the data and provenance here before this material is quite ready for publication.

We thank the review for believing that our work will be of value to the community and for their very constructive criticism. We have taken all of their comments on board and have tried to address them as thoroughly as we can. In addition to changes and additions to the main text and supporting information (SI), we have uploaded the benchmark data set to Github (https://github.com/schrodinger/public_binding_free_energy_benchmark), and provided more details and metadata that are more in line with FAIR data standards.

Major comments:

- In terms of selecting systems for benchmarking FEP, or in categorizing systems on which FEP is benchmarked, the authors really ought to make contact with the "Best Practices" work by Hahn et al., (LiveCoMS, <https://livecomsjournal.org/index.php/livecoms/article/view/v4i1e1497>) which is intended to provide community standards for FEP benchmarks. The paper is CITED here (with missing details) but there is no mention of it or discussion of it, which seems strange given that it seems to represent the current community consensus on the topic. Even if the authors disagree, they should discuss it and explain where they disagree/why. Additionally, that study (e.g. Table 1) looks at quality metrics concerning choice of protein structure, gives examples concerning common problems with protein structures, and gives a checklist for assessing whether problems with structures would be likely to affect results. These issues should certainly be considered and reported for the targets considered here.

We agree that the manuscript did not properly address the context of the field, and omitted important details of other benchmark data sets, especially that of Hahn et al. The work of Hahn et al. is now discussed and cited throughout the revised manuscript; these sections are shown in blue text below in our subsequent responses below.

- Relatedly, careful reporting of what PDB codes are used for studies is essential, but these are not always listed; for example, the SI section on Jnk1 does not report what PDB code was used. Some prior work on Jnk1 had used code 2GMX, which is a structure with considerable problems. What was used here? Somewhat relatedly, the 5HNB structure is used for the study here, but this was called out by the benchmarking study (noted above) as being of lower quality as some other structures such as 5XS2, 4CRL and others. Given this, the authors had to build in unresolved loops as they note -- "The latter two missing loops may affect the accuracy of FEP..." Why were

the better structures recommended by the earlier work not used? (This may hold true across other targets as well; CDK8 was only the first example I checked against Table 1 in the benchmark study; I suggest the authors carefully go through all systems in that table).

We regret that the PDB codes were not reported in the original manuscript. In the updated SI, we now list the PDB codes of the proteins in the tables that summarize FEP+ accuracy in each subgroup. PDB codes can also be found in the metadata that accompanies the benchmark in the Github repository. We thank the reviewer for pointing this out, as this data's omission was clearly the reason why they believed we used PDB code 5HNB as the crystal structure for CDK8. We in fact used the higher quality structure 5CEI, for reasons we discussed in the first paragraph on page 7 of the SI. The discussion of the different crystal structure and its validation was over 300 words long and appeared in the paragraph after the one with the sentence "The latter two missing loops may affect the accuracy of FEP..." and can be found on page 7 of the SI.

We note that using a different protein structure is not trivial and often requires significant validation. We discussed extensively in the SI that, we used different or modified starting structures for a number of different systems, including CKD8 and SYK (from the Merck set), HSP90 and JAK2 (from the fragment set), and explored different protonation state assignments with A2A (from the GPCR set) and, in the case of one of the Janssen BACE1 proteins (the P3 series), the effect of alternate side chain rotamers on the pKa of a buried residue. As we hope our detailed SI communicates, protein structure selection and preparation is an evolving science and the best model can change as new methods or crystal structures become available.

With regards to the JNK1 crystal structure (PDB code 2GMX), we agree that it is of a lower standard than others in the benchmark data set (it has for instance, a reported resolution of 3.5 Å). Since 2GMX is the only structure co-crystallized with a ligand in the chemical series being modeled, we did not undertake an exploration of other crystal structures although some other crystal structures might have higher resolution. We also note that this same crystal structure was used in the validation of the OpenFF's 2.0 (Sage) force field (reference 38 in the main text) which uses the benchmark set by Hahn et al., so at the very least, our use of this structure is in line with the rest of the field.

To communicate the efforts by Hahn et al in this area as well as to address the quality of the JNK1 structure, on page 5 of the main text, we added the statement:

"We note that the work by Hahn et al.\cite{Hahn2022} lists alternative crystal structures for many of the systems used in this benchmark which could be considered in future development of our benchmark set."

And on page 30 of the SI, we have added the statement:

"We note that the crystal structure used as the basis of the protein model has resolution of 3.5 {AA} that is higher than most other crystal structures that are used in this benchmark. The poorer quality of this crystal structure was highlighted by Hahn et al., who suggested alternative crystal structures. Future developments of this system will explore alternate, higher quality crystal structures."

- The same benchmarking study recommends minimum standards for dataset size and dynamic range for benchmarking FEP; again, those standards seem to be ignored here (e.g. for Ftase and

BACE in Table 2 have particularly low counts; I haven't checked dynamic range yet but the authors likely should check)

The restriction to having data sets with a minimum number of compounds and dynamic range of experimental free energy is a particular choice by Hahn et al. that we respect but do not believe is appropriate in our study. We have added a discussion of this important point on page 4 of the main text:

Unlike Hahn et al., we did not omit chemical series on the grounds of having too few compounds or too small a dynamic range of the experimental binding free energies. While having such requirements leads to tighter prediction error and correlation metric confidence intervals on *individual* chemical series, we are primarily concerned with the *aggregated* performance of FEP on the whole data set. Metrics such as overall RMSE or Kendall's tau can be calculated with high confidence even if the data sets they aggregate over are small or have a low dynamics range. We were motivated to include some of the smaller data sets because they covered specific chemical types or transformations that are not present in other larger benchmark data sets. Examples of these data sets include those focused on macrocycles, charge changing transformations, and buried water displacement.

Elsewhere in the main text, we have added extra statements that clarify that one of the primary aims of our work was to have a benchmark data set that covered as much chemical and transformation diversity as possible. The omission of some of these smaller data sets would have meant that some chemotypes would not have been covered. Also, we were more interested in the aggregated performance of FEP on as large a data set as possible, and thus were motivated to include smaller data sets. On page 4 of the main text, for instance, we now write:

Two aims for this benchmark data set were to include as many ligands as possible and to cover the range targets and ligand perturbations that occur in drug discovery, such as charge changing and/or buried water displacing perturbations. By meeting these aims, we hoped that the resultant benchmark would provide a thorough test for current and future FEP methodologies. While the majority of the systems in this benchmark come from previous FEP studies, additional data sets with well resolved protein structures and ligand binding affinities that were encountered during the assembly of this data set were also included.

In the new metadata of the SI, we also list the number of compounds and dynamic range of each data set such that future users of this benchmark can decide whether they would like to exclude certain data sets.

- The same study provided a python package to report standardized analysis statistics for exactly this type of benchmark; again, the authors should either use the package, report the same statistics (and provide their own implementation), or clearly justify why they are not. For future work, it's essential that benchmarking studies like this be able to be compared, and if each study implements its own analysis and doesn't release the code, it becomes extremely difficult to compare, hence the need for standardization.

We completely agree with the reviewer that a set of community agreed accuracy metrics will be greatly beneficial for comparing different free energy prediction methods. We discuss in detail in section 2.2.2 (starting on page 5) the reasons for our choice of metrics, namely the edgewise errors, pairwise errors of the relative binding affinity predictions, and the R-squared and Kendall tau of the (inferred) absolute binding free energy predictions. The edgewise errors and Kendall tau metrics are, as we understand, completely congruent with the metrics suggested by Hahn et al. and in the OpenFFv 2.0 (Sage)

publications. To aid comparisons, we also report errors as both root-mean-squared errors and mean unsigned errors.

As we discuss on page 5, we believe that the pairwise error is a more robust measure of performance than edgewise errors. We have added a new statement that acknowledges the use of this metric by Schindler et al. at the top of page 6.

We note that this weighting scheme, along with the use of pairwise errors, was used in the FEP assessment by Schindler et al.

We hope that our discussion of the benefits of the pairwise error, along with its presentation alongside edgewise errors forms part of the discussion that leads towards a consensus on what metrics are most appropriate for free energy calculations.

Also at the top of page 6, we clarify we do not use the correlation statistics of the relative binding free energy predictions:

We do not report the correlation statistics for the predicted $\Delta\Delta G$ s because, as illustrated by Hahn et al, these statistics are dependent on the sign of the arbitrary sign of the $\Delta\Delta G$ s.

In the new Github repository, we have included the (python-based) code and analysis scripts that we used to produce the summary statistics and the main figures of our paper. To help users repeat our analysis, we have included the predictions from FEP+ in two forms. The first is in the format that is native to FEP+ (a graph-based object in .fmp format). The second format is as human readable .csv files, which will allow practitioners who do not have an FEP+ license to repeat our statistical analysis.

- Because benchmark results are so sensitive to setup details, it's essential that the inputs for the reported calculations be provided in a useful and reusable way, with clear documentation, so that others can build on this work. While some, or much of, the data seems to be provided it is not completely clear how to use it. For example, the provided ZIP file contains a subdirectory on `fep_benchmark_data`, which has no documentation (e.g. README.md files) or metadata. Guessing, it seems safe to assume that the subdirectories there are organized by protein-ligand set as laid out in the SI, but the names don't really correspond, e.g. there's a "jacs_set" directory which contains ... what? And what's "misc"? Or "waterset"? This material should be compiled in a way that it clearly corresponds to what is laid out in the paper AND has metadata explaining provenance, otherwise it is both (a) irreproducible, and (b) very hard for anyone else to reuse for anything. I would have similar concerns with the experimental survey data, though I've not looked at it for as long. (However, I note that much of the data there is .csv files without column headers which have names like "aaron2010_spr_itc.csv"; presumably these need (a) column headers, and (b) metadata properly citing the papers, explaining details of where the data came from, and detailing how it is laid out. The authors should consult the FAIR data standards (e.g. https://en.wikipedia.org/wiki/FAIR_data) and at least attempt to adhere to SOME of these standards; at present, I am not convinced even a human expert could adequately interpret the provided data package. (As a concrete example of this concern, if I wanted to reproduce Figure 3 as the first step of extending this analysis, exactly what data should I analyze?)

We thank the review for this clear and important critique. As we have mentioned above, we have improved the FEP and experimental benchmark data set and have made it freely available on github (https://github.com/schrodinger/public_binding_free_energy_benchmark). This repository is a much more

detailed and informative version of the SI data we originally submitted with this manuscript. Following the reviewer's comment, we have endeavored to make the data in the repository follow more closely with the FAIR data standards. New improvements to the supporting data include, but are not limited to:

- Extra documentation in the form of README files spread throughout the repository. These readme files discuss the contents of the repository as well as its usage.
- Metadata files for the FEP benchmark, which includes protein names, PDB codes, the number of nodes (e.g. ligands) in each series, the dynamic range of the experimental data and the DOI of the original citation. The abbreviated group names that the reviewer referred to are now also linked to the more interpretable group name as given the SI.
- A metadata file for the publicly accessible experimental benchmark. This metadata links the file names (e.g. "aaron2010_spr_itc.csv") to the protein, the publication DOI, where in the publication the data was extracted, and the name and type of assays that are in the file.
- Headers have been added to the experimental .csv data files. These headers state the assay type and units.

We have spent a considerable amount of time improving the quality of this supporting data. As per the reviewers suggestions, we have endeavored to make the data in the repository follow more closely with the FAIR data standards. Being a publicly accessible repository on Github..

- The authors should consider what they would want to happen to this dataset of curation/correction is needed, e.g. if someone finds that one of the ligands used in the study is not the one which was actually studied experimentally. Do they wish to submit an erratum to correct the structure? (Or, likewise, if there is a typo in one of the experimental values given in the SI data package). One approach to this problem would be to provide the data in a GitHub repository under version control rather than in files buried in the SI of a paper (where they can only be changed/corrected by submitting an erratum). This would allow the data used in the paper to correspond to one specific version of the repository, but still allow room for further updates and corrections.

This is an excellent point, which is why we have made a greatly improved version of the FEP benchmark and experimental survey available on Github (https://github.com/schrodinger/public_binding_free_energy_benchmark). While we have what we believe to be substantial improvements to the supporting data (as discussed above), we realize that there may still be room for improvements. We, like the reviewer, think that Github is an excellent repository for the data as improvements can be added in a continuous fashion and users will be free to raise issues.

- There seems to be no discussion, either in the paper or in the SI, of what is provided in the supporting zip file, so a reader not looking for it might not be even aware it is present.

We have added a new section in the main text after the Acknowledgements but before the Appendix, starting on page 11, that summarizes the contents of the SI.

While some of the above remarks concern dataset organization and quality and might be seen as a relatively minor issue, this paper presents the benchmark data set as a core deliverable/outcome, e.g. "We have assembled ... the largest benchmark data set to date...". If this is to serve as a benchmark data set, it should be better organized, and have appropriate metadata and provenance information, and the authors should better document the choices made in selecting their specific protein structures, etc., especially in view of the prior work on best practices for benchmarking these methods. There are critical choices which should also be

documented, e.g. how were any missing residues or loops handled, were internal waters retained or removed, what protonation states (for proteins and ligands) were chosen and why, how were ligands posed, etc. Some of these issues are discussed somewhat in the SI, but not completely, e.g. the Cmet section says that 8 ligands were re-docked, but doesn't list WHICH 8 ligands were re-docked. The Syk section notes that "several of the ligands were redocked", but not which.

We hope the reviewer agrees that substantive additions that we have made to the supporting data and its availability on GitHub, which we have discussed above, addresses their valid concerns regarding the organization of the data.

With regards to our general system setup methodology, we discuss our general procedure in the Simulation Details section (2.2.1) on page 5 of the main text. As we state there, "All crystallographic water molecules were retained and missing side chains or loops were added with Prime. Protonation state assignment was carried out with PROPKA".

We documented any deviation from the general setup in the SI, as well as documented and justified and validated as best as we could any modifications we made to systems relative to their original sources. We believe the scale and scope of the SI, the summary in section 3.1 and figure 1 of the main text exemplify the importance we place on system setup and how critical we believe these details are. Systems where protonation states of the ligand or protein were discussed in detail include TNKS2, HNE, and BACE1 P3 from the Janssen set to name a few. Cases where we investigated binding mode sensitivity include HIF2 α , and A2A. Protein structure changes were investigated in a number of systems, including JAK2 and CDK8. Cases where we discussed the use of "custom cores" include FXa and P2Y1. We believe that the SI already provides a great deal of details regarding system preparation and we have done our best to document the changes we made.

We thank the reviewer for pointing out two examples (Cmet and SYK) where we did not explicitly state which ligands were redocked. This was an oversight and tables S9 and S12 have been added to document these details. For JAK2 in the fragment group, we clarified that all ligands were redocked (page 14 of the SI).

Indeed, similar issues have affected reuse of data from the 2015 "JACS set" paper, also from Schrodinger. Structures were deposited, and have been broadly reused with some benefit to the community, but also have left many questions unanswered. Some systems retained buried waters, others removed them, some built in missing loops, others didn't, and many choices were left undocumented with answers spreading only through the grapevine by way of human experts who have spent a great deal of time with the data. In the present study, the authors have an opportunity to recognize that (as they likely hope) this data will be broadly reused by the community and come up with a better plan for documenting the protocols used in preparing it and addressing any questions and concerns that may arise. Again, this may argue towards making the data available in a GitHub repository where subsequent questions can be answered (e.g. if someone has a specific question about a particular choice the authors forgot to document, it can be answered once-for-all in the relevant GitHub repository) and corrections made if necessary. I don't think the authors have to be compelled to do this, but certainly something better than the current data (lack of) structure is warranted.

We completely agree with the reviewer on the need for clearly documented protocols as well as the need for accessible and accountable data. As we discussed above, we have significantly improved the documentation and metadata of the FEP benchmark and experimental survey and have made it freely

accessible on Github. This will allow, for instance, future users of the benchmark to raise and discuss possible issues of the benchmark and will facilitate its continuous development.

We would again like to draw attention to the extensive detail we have provided in the SI whenever we investigated and developed the structural inputs from the FEP benchmark's data sources. We hope that this SI, together with the better documented and accessible accompanying data will help maximize the utility of our work for the community.

Minor remarks:

- The references are very heavy on Schrodinger work; perhaps good to add more non-Schrodinger citations

In the course of addressing all 3 reviewers' comments, we have added numerous additional citations, including, but not limited to

- As the first reference of alchemical binding free energy methods: Mey, A. S. J. S., Allen, B. K., Bruce Macdonald, H. E., Chodera, J. D., Hahn, D. F., Kuhn, M., Michel, J., Mobley, D. L., Naden, L. N., Prasad, S., Rizzi, A., Scheen, J., Shirts, M. R., Tresadern, G., & Xu, H. (2020). Best Practices for Alchemical Free Energy Calculations [Article v1.0]. *Living Journal of Computational Molecular Science*, 2(1). <https://doi.org/10.33011/livecoms.2.1.18378>
- To the list of packages that can perform alchemical free energy methods, we've added the latest paper from AMBER in addition to its previous ones: Ganguly, A., Tsai, H.-C., Fernández-Pendás, M., Lee, T.-S., Giese, T. J., & York, D. M. (2022). AMBER Drug Discovery Boost Tools: Automated Workflow for Production Free-Energy Simulation Setup and Analysis (ProfESSA). *Journal of Chemical Information and Modeling*, 62(23), 6069–6083. <https://doi.org/10.1021/acs.jcim.2c00879>
- Also to the list of alchemical packages, we added another alchemical GROMACS paper: Gapsys, V., Hahn, D. F., Tresadern, G., Mobley, D. L., Rampp, M., & de Groot, B. L. (2022). Pre-Exascale Computing of Protein–Ligand Binding Free Energies with Open Source Software for Drug Design. *Journal of Chemical Information and Modeling*, 62(5), 1172–1177. <https://doi.org/10.1021/acs.jcim.1c01445>
- To the list of citations of where FEP was applied to drug discovery projects on page 2: Jama, M., Ahmed, M., Jutla, A., Wiethan, C., Kumar, J., Moon, T. C., West, F., Overduin, M., & Barakat, K. H. (2023). Discovery of allosteric SHP2 inhibitors through ensemble-based consensus molecular docking, endpoint and absolute binding free energy calculations. *Computers in Biology and Medicine*, 152, 106442. <https://doi.org/10.1016/j.compbio.2022.106442>
- The Open Force Field paper for version 2 (Sage) that includes the benchmark assessment of Hahn et al.: Boothroyd, S., Kumar Behara, P., Madin, O. C., Hahn, D. F., Jang, H., Gapsys, V., Wagner, J. R., Horton, J. T., Dotson, D. L., Thompson, M. W., Maat, J., Gokey, T., Wang, L.-P., Cole, D. J., Gilson, M. K., Chodera, J. D., Bayly, C. I., Shirts, M. R., & Mobley, D. L. (2023). *Development and Benchmarking of Open Force Field 2.0.0-the Sage Small Molecule Force Field*. <https://doi.org/10.26434/chemrxiv-2022-n2z1c-v2>

- p2 concerning domain of applicability in benchmark sets: What is the evidence for this assertion (that the domain of applicability is not covered adequately in present sets)? The paper also never seems to revisit the idea of domain of applicability later, either, so it is unclear that the present study does a better job covering the full domain of applicability. (The Discussion notes some

reasons why the data collected here might be biased towards particularly well-performing cases, which would suggest that this set does not yet probe the edges of the domain of applicability.)

We thank the reviewer for bringing this to our attention. Improving the domain of applicability of current benchmark datasets was one of the motivating factors of this work. To address this comment, we have added the following paragraph in the introduction (page 2):

The two largest protein-ligand benchmark data sets for FEP are the set assembled for the OPLS4 force field by Lu et al. (with a total of 512 protein-ligand pairs)^{Lu2021} and the set by Hahn et al. (with 599 protein-ligand pairs); the latter of which was designed to be the community standard and has been used in the validation of the Open Force Field 2.0.0. Unlike data sets such as PDBbind, these data sets do not contain only experimentally determined structures, and instead consist of congeneric series of ligands where all ligand binding modes and protonation states have been modeled. While these benchmarks have proved to be very useful they currently do not cover the domain of applicability of FEP, lacking, for instance, any membrane proteins, scaffold hopping transformations, and macrocyclic transformations to name a few deficiencies. Although the OPLS4 benchmark set includes charge-charging and buried water displacing transforms, the benchmark set by Hahn et al. does not. With the benchmark set by Hahn et al., the omission of certain data sets may be by design, as only data sets that met certain quality standards were included.

We have also have expanded on the follow-up discussion in the Discussion section:

FEP is most useful when it can be applied to the kinds of designs that are made by medicinal chemists, which is why we have included a wide variety of transformation types in our benchmark, such as those involving fragments, scaffold-hopping, charge-change, macrocyclization, and water displacement. The range of systems and transformations contained in our benchmark covers a wider domain of applicability than previous benchmark sets, namely the OPLS4 set and the set by Hahn et al.. Our FEP benchmark set also contains over twice as many ligands as either of these data sets. With these considerations in mind, we hope our benchmark dataset provides the most comprehensive test of an FEP method to date. It is also our hope that FEP benchmark datasets will continue to grow in size, ligand diversity, and target coverage. The benchmark data set we present here is a step forward in this respect, although there remain areas for improvement. The number of membrane proteins in this benchmark set could be increased in future iterations to better reflect the distribution of drug targets, but we note that the aforementioned prior benchmark sets did not contain any membrane proteins. Other types of ligand transformations, such as those involving transition-metal complexes which are actively being developed were omitted from this dataset and should be included in future efforts. Although we reviewed the quality of many of the protein and ligand structures in this work, this aspect of the data is an evolutionary process and the structures should remain under review in future versions of this benchmark.

- There are missing references, e.g. p9 bottom refers to "Schindler et al." but no reference is given and a search for the name "Schindler" only turns up reference 32, which is not the intended paper. (The authors may have meant the recent large scale benchmark study led by Schindler, which certainly should be cited here.)

In that section we did indeed mean the large scale benchmark work and have added the citation to that section.

- page 2 near bottom, notes studies across within-team vs across-team assays. I believe there is also work which separates out the effects of "fresh solutions" vs "reused solutions", e.g. a same-team experiment could use a fresh prep of all the solutions or reuse the solutions. I know this had been a major point of some of John Chodera's work on ITC, e.g. that teams don't adequately account for concentration uncertainty when reporting experimental uncertainties. It would be nice to see some discussion of the potential sources of uncertainty here.

- Relatedly, probably the prior experimental ITC cross-comparison (looking at ITC done on the same systems across labs, and the reported free energy, enthalpy and entropy) should be discussed here. I don't know the reference, but I know it's discussed in some of Chodera's work on ITC.

The sources of experimental uncertainty in binding affinity measurements are numerous and fascinating and a whole review paper is needed to do the topic justice. Although the focus of this manuscript is on the quantification of accuracy and reproducibility, we have added a new paragraph in the introduction that briefly touches on this subject.

We are familiar with John Chodera's work on the sources of error in ITC measurements and have now cited two of his papers on the topic, but were unable to find the citation that discussed "fresh solutions vs reuses solutions". We had already cited the ITC cross comparison study in the introduction.

There are numerous factors that drive the observed differences in measured binding free affinities between laboratories. These can range from concentration errors in, for instance, the reagents used in isothermal titration calorimetry experiments, to the difference in material of the assay containers; one study reported that a particular compound was absorbed by glass and not plastic, which artificially raised the apparent K_i of one compound. Significant differences in measured binding affinities can also occur when assays are repeated with alternate instruments or when data is reanalyzed with different software. Data fitting methods can be made more robust when using analysis methods that explicitly model uncertainties in the experiment.

Reviewer #2 (Remarks to the Author):

Ross et al. present an investigation into accuracy of experimental binding affinity determination and compare this to an evaluation of predicted binding affinities using free energy calculations. They collect a large dataset for free energy calculations from the literature and report performance of the FEP+ method on this based on optimized input structures.

The investigation into experimental accuracy for binding/functional assays is a good extension of previous work by Brown et al. and Kramer et al. and will be very useful for the field. The compilation of the dataset for evaluating free energy calculations fits well with other recent efforts in the community and will be of interest to many scientists in this area. Similarly the case by studies provide interesting learnings on best practices. I recommend the article for publication with minor suggestions for modifications.

Comments:

- Is there a plan to include this new dataset/parts of it into the Protein-Ligand benchmark at <https://github.com/openforcefield/protein-ligand-benchmark?> There are already efforts ongoing to rework some of the datasets that were included here.

We have not yet made contact with the authors of the paper mentioned, but we hope that the data we provide and the descriptions of each set's development in the SI will be of use to future benchmarking efforts.

That said, we note that the work by Hahn et al places restrictions on what data sets can be included in their benchmark set, based on the number of compounds in the series and the dynamic range of the experimental data. We do not believe their restrictions are appropriate for our benchmark data set given that many important chemical types and transformations data sets are small and would be excluded with such restrictions. We have clarified our position in the main text on page 4 with the new statement:

Unlike Hahn et al., we did not omit chemical series on the grounds of having too few compounds or too small a dynamic range of the experimental binding free energies. While having such requirements leads to tighter prediction error and correlation metric confidence intervals on *individual* chemical series, we are primarily concerned with the *aggregated* performance of FEP on the whole data set. Metrics such as overall RMSE or Kendall's tau can be calculated with high confidence even if the data sets they aggregate over are small or have a low dynamics range. We were motivated to include some of the smaller data sets because they covered specific chemical types or transformations that are not present in other larger benchmark data sets. Examples of these data sets include those focused on macrocycles, charge changing transformations, and buried water displacement.

- Could the authors add the error bars for the FEP predictions and potentially if reported for the experimental data in Figure 2?

- Please add the confidence intervals to the statistics in Figure 2.

We have completed both of these suggestions. Unfortunately, experimental uncertainties were not reported for both of the FAK data sources and the FXa series.

- Reference 32 is incomplete.

This reference has now been fixed.

- Reference to Koehler et al. missing in SI page 5.

This reference has been added.

- The individual results for all datasets are scattered across the SI. Could the authors collect these in a single table or provide a single table in addition to the smaller tables?

We have added a new table in the SI (table S6) that lists the results for each grouped data set in the FEP benchmark.

The results for each individual chemical series have been collected and added to the expanded and better documented supporting data that is now available on Github (https://github.com/schrodinger/public_binding_free_energy_benchmark). There, the results for each individual set can be found in the directory 21_4_results/summary_statistics. The benefit of placing all of the individual results there (in addition to what is found in the SI) is that the accuracy on each data set is reported with more accuracy metrics than could fit on a line in the SI.

Reviewer 3

The work collects a large number of binding affinity data measured in experiments and evaluates the uncertainty in measured free energy differences from cross-experiment comparison. Further, another large set of binding affinities is estimated by means of free energy calculations. Comparing calculation results to the experimental measurement showcases that calculation reaches nearly the experimental error. Overall, this is a very good work: well structured, executed and written. I have three main points that in my opinion would still need to be addressed:

We thank the reviewer for their positive review and very helpful comments.

• Given that the work already provides comparisons of experiment_vs_experiment and computation_vs_experiment, it is natural to be interested in the error between two computed results, i.e. computation_vs_computation. The authors have now calculated free energy differences following their best practice in structure and simulation preparation. However, most (maybe all?) of those values have already been evaluated previously. Would it be reasonable to consider the previous calculations as independent experiments, where researchers might have followed a slightly different protocol and assess what is the level of reproducibility for calculated $\Delta\Delta G$ s?

The reviewer has raised a very interesting point that is worth fully considering. The quality and accuracy of a (relative or absolute) binding free energy predictions are dependent on the accuracy of the force field, the degree of sampling, and the reliability of the workflows and software used to produce the predictions. At present, there is a wide range of quality and accuracy from the different software packages and force fields. Popular protein and small force fields, such as OPLS, AMBER and GAFF, and the Open force field, are regularly updated to what are designed to be more accurate and comprehensive versions. Additionally sampling methods are continuously improved, such as the enhanced water sampling method that FEP+ employs.

Given this varied and evolving landscape of models, the value of aggregating all the prior estimates in a single uncertainty measure is unclear to us. For instance, for perturbations that displace buried water molecules, there are a number of documented systems where methods that enhance the sampling of water are known to produce very different (and more accurate) predictions compared to those that do not. For these types of perturbations, would the difference between the predictions of one method with enhanced sampling and one without be indicative of predictive uncertainty? We would argue in the negative, as one set of the predictions (the one that did not have enhanced sampling of water) would be known to be inadequate for the particular perturbation. One can make similar arguments regarding force fields, as force field updates often tackle known issues of the previous version.

The FEP+ method and OPLS4 force field are highly likely to be the most accurate FEP methods to date. The very recent publication of version 2 (Sage) of the Open Force Field compared the accuracy of FEP+ with OPLS3e (the version preceding OPLS4) against GAFF2, OpenFF v1 and OpenFF v2 on the protein-ligand data set of Hahn et al. They found that, on average, FEP+ had a lower RMSE and a high Kendall Tau than the other methods. Merging predictions of FEP+ with other methods at this point in time would be likely to degrade accuracy and not be representative of the state of the art.

Quantifying the true uncertainty of alchemical binding free energy methods is unfortunately quite difficult and most methods (which are usually based on sampling noise) significantly underestimate the uncertainty.

• It would be interesting to also provide the average uncertainty reported with the experimental measurement itself (understandably, not every published measurement has an associated uncertainty, but certainly a fraction of them do report it). It would be great to know how much of the experiment_vs_experiment error can be accounted for by the uncertainties reported by the experiments themselves. And similarly, FEP+ also reports an error associated to each individual free energy calculation. How does this uncertainty compare to the difference observed from computation_vs_computation?

We thank the reviewer for this interesting and important suggestion. We have added a new experimental survey to the SI on page 5, section 1.1 called “Experimental uncertainty from repeating the same assay”. In this section, we compiled reported standard deviations from 15 different assays and calculated the overall root-mean-square standard deviation. This estimate of the uncertainty from repeating an assay helps place the reproducibility estimate in table 3 of the main text in context. We have added the following discussion in the main text (page 10):

The primary interest of our experimental survey is to quantify the reproducibility of experimental binding affinities, which we have approached by quantifying the difference of (relative) binding affinity measurements from *different* experiments. Nevertheless, it is of interest to place the experimental RMSEs in context of the uncertainty that arises from multiple repeats of the *same* experiment. In the table S5 of the SI, we collect the reported standard deviations from repeats from a total of 350 compounds spread over 15 experimental assays. The overall root-mean-square of these reported standard deviations is 0.23 [0.18, 0.33] kcal/mol, where the square brackets denote 95% confidence intervals that have been calculated with bootstrap sampling over the different assays. The uncertainty from the repeats of a single experiment contributes to RMSE when comparing different experiments. Assuming the repeatability error we have calculated is Gaussian distributed and applies to all experiments, the RMSE from taking the difference from two measurements is approximately 0.33 kcal/mol (from the square root of 2 times 0.23 kcal/mol). This value is roughly a third of the 0.91 kcal/mol reproducibility RMSE from table 2, which implies that two thirds of the reproducibility RMSE comes from intrinsic differences from the different experiments.

With regards to uncertainty to the computed FEP estimates – following the advice of reviewer 2 – we added confidence intervals to figure 2 from both experiment (where available) and FEP+. The uncertainty estimates from FEP+ were calculated using the statistical estimate from the cycle closure algorithm, which, as the figure shows, are on the order of 0.1 kcal/mol. The estimated statistical uncertainty of FEP+ is much smaller than the RMSE with respect to experiment, which is likely dominated by force field errors and barriers in the configuration sampling.

• The authors did a very thorough job in going through a large number of previously assembled FEP datasets identifying the outliers and fixing or discarding them. Such a retrospective analysis of the outlying calculation results naturally lowers the computation_vs_experiment RMSE, but also makes it not fully representative of the deviation that could be expected in a prospective prediction. I think it would be important to emphasize this point, as this RMSE will not necessarily directly transfer to the prospective drug design projects. In addition, would such a rigorous outlier analysis also reduce the experiment_vs_experiment RMSE? Currently, the authors rather

excluded very similar experimental setups (as described in the SI page 3), so is the experiment_vs_experiment RMSE somewhat skewed towards the larger values?

We wholeheartedly agree with the reviewer on this important point. Indeed, in the discussion section, we had the paragraph that began with the sentences: “For our FEP benchmark, we primarily collected data sets from previous publications on FEP. This potentially biases our estimates of FEP error as the data from prior publications are more likely to contain systems where a particular FEP method appeared at least satisfactory in terms of accuracy.”. To further emphasize this point, we have added the additional follow on sentence in that paragraph:

It should also be noted that our estimate for the error of FEP+ with respect to experiment was estimated retrospectively and that reported errors are typically higher in prospective applications.

In addition, would such a rigorous outlier analysis also reduce the experiment_vs_experiment RMSE? Currently, the authors rather excluded very similar experimental setups (as described in the SI page 3), so is the experiment_vs_experiment RMSE somewhat skewed towards the larger values?

The purpose for performing the experimental error survey in conjunction with the FEP+ benchmark was to quantify the highest possible accuracy we could ever expect from a computational method. In the ideal case, comparing a computational binding affinity prediction to an experimental measurement would be like comparing two completely different experimental measurements. Every experimental affinity assay involves using a model of some kind, from the equations used in the data analysis (e.g. once derived from assuming a high ligand concentration), to the protein model (e.g. assuming that the immobilized protein acts like a free protein in SPR), to the assumed equivalence of K_{is} and K_{ds} (e.g. a ligand may not bind at the orthosteric site and not affect the enzymatic activity.), etc. We believe that the only way to interrogate these experimental models and assumptions is to compare very different assays.

Thanks to the reviewer’s previous suggestion, we have now added a section in the SI (section 1.1 page 5) and a new paragraph has been added to the main text (starting on page 9) about the uncertainty that comes from repeating a measurement multiple times. Hopefully, this will place our reproducibility error in context as the largest possible uncertainty that exists on the ladder of experimental uncertainties.

Below I also collected several minor questions and comments. After addressing these questions and comments I would recommend the article for publication in Communications Chemistry.

Introduction

On page 2 it is mentioned that a number of programs can perform FEP. Among them Amber based calculations are referenced. To my knowledge Amber uses TI rather than FEP.

In our experience, the term FEP has become the all encompassing term for alchemical free energies, no matter the type. We have clarified our position on the 2nd paragraph of the introduction with the statement:

The term free energy perturbation (FEP) used to refer to a particular class of alchemical method but has in recent years been applied to alchemical binding free energy methods more generally; henceforth, we will refer to all alchemical binding free energy methods, including thermodynamic integration, as FEP.

Prior to this statement, we have also replaced the two instances of “FEP” or “free energy perturbation” to “alchemical free energy calculations”.

Methods

For self-consistency of the manuscript it would be useful to have a bit more elaborate description of the simulation methods. In particular, charge changing modifications are known to be challenging. Which strategy did the authors use: was the overall neutral charge of the box conserved during the simulations or was a post-hoc correction applied?

We have added significantly more detail on the simulation methodologies that we used on page 5, section 2.2.1. This additions include a statement on the charge changing transformations:

For alchemical transformations that changed the charge of the ligands, the total charge of the simulation was kept constant by transmuting an ion either to water or vice versa, depending on the charge differences using the scheme previously described.

And we have cited the paper that described FEP+'s method for these cases.

Using FEP to resolve ambiguities in the structural inputs

To include multiple rotamers into a perturbation map (and later apply equation 4) the necessary condition is for the rotamers not to interconvert between the states during the alchemical calculation. How did the authors ensure that: or was it just an observation after the simulation? Was the REST not sufficient to lower the transition barriers for the rotamer interconversion?

On page 6 we have added the text which should address the reviewer's questions:

In cases such as this, replica exchange solute tempering was not sufficient to lower the sampling barriers enough to facilitate complete rotamer sampling; extra rotamers were added to the perturbation map when poor rotamer sampling was observed in prior simulations using the automatically generated FEP+ analysis panel.

For the PTP1B case, what would be the interpretation for such a low cysteine pKa in proximity to a carboxyl group on the ligand? Is it not unfavourable to have two partial negative charges close to one another when cysteine is deprotonated?

In the main text (page 6) we have added some additional context to this case:

In this system, the sulfur atom of a cysteine residue sits within a bowl of backbone NH groups but it is also in close proximity to the carboxylic acid group of the ligands; this mixed electrostatic environment makes the determination of the cysteine protonation state nontrivial.

In figure 1, the NH groups that make the negatively charged state of the cysteine are not shown, which we now understand makes this result appear confusing. We have also added the sentence to the caption of figure 1:

In PTP1B, the sulfur atom is in close proximity to a number of backbone NH groups (not shown) which are likely to be the the main drivers of the predicted negatively charged protonation state

For the CK2 cyclization example, from a practical point of view, how one should identify in advance which barriers will not be crossed and thus need a special treatment? How does one in practice change atom mapping in FEP+?

For the CK2 case, the lack of torsion sampling was noticed by visual inspection in a prior FEP calculation and manually changing the atom mapping within the FEP+ panel. (This is achieved by choosing a “custom core” outside of which all atoms are made to be in the perturbation region.)

Two main areas of approach that come to mind for automating the sampling benefits that required manual intervention from ourselves. The first main area of approach would be an automatic post simulation analysis that scans the torsions and detects where there was a lack of sampling and the automatic generation of a new atom mapping. The second main area of approach would be to *always* scale the torsion potentials to zero in the intermediate lambda windows for macrocycles. The latter class of approach is simpler to implement than the former, and is being considered in newer versions of FEP+.

We have added a short statement to this effect in the main text on page 8:

While the changes described here came from visual inspection and manual intervention, an automated approach for these kinds of perturbations would be preferable.

In general, more work is needed to automate and generalize the improvements to the setup of FEP that we have used in this work.

Figure 2

It would be good to specify that ΔG was offset with the experimental mean of the same experiment to which FEP+ is compared to. Otherwise, it may be ambiguous which experimental set to choose, e.g. for the galectin case taking ITC mean would offset the values with respect to the biophysical FP experiment.

This correction has been made.

Figure 3

What do the boxes and whiskers represent? Are the whiskers 95% CI for the whole pairwise RMSEs (seems somewhat confusing as all the visualized points fall within the whisker range for FEP+)?

We have added the following sentence to the caption:

In the boxplots on the left, the boxes represent the 25th and 75th percentiles and the dark line represents the median. The whiskers extend to a maximum of 1.5 times the interquartile range.

For the histograms Figure 3 (right panel) would it make sense to note whether the pairwise $\Delta\Delta G$ in both directions were considered? In the earlier RMSE calculations, as I understand $N \times (N - 1) / 2$ pairs were considered, but for this figure the signed errors were explored, so probably all $N \times (N - 1)$ were used?.

The reviewer is quite correct. In the caption of figure 3, we have added the sentence:

The histogram was symmetrized about the $x=0$ line in the sense that all $N \times (N - 1)$ pairs of compounds were used.

Also, the text on page 8 explains: “the percentages one would expect from a Gaussian distribution with the same pairwise RMSE”. Is it rather meant that a Gaussian with the mean and variance of the $\Delta\Delta G$ error distribution from Figure 3 was considered?

This was poorly phrased and has been updated to:

...the percentages one would expect from a Gaussian distribution that has a standard deviation equal to the pairwise RMSE of FEP+.

Appendix

The first sentence in A.1 is confusing. It states that equation (4) is for R-groups with rotational symmetry, but equation (4) is general and can be applied to any conformations. Probably it is meant that a special case of R-groups with rotational symmetry is discussed in A.1, but then some clearer wording would help.

We thank the reviewer for their suggestion. We have changed the leading sentences of section A.1 (page 13) to

Equation 4 is general and can be applied to multiple conformations of any type. One type of conformational degree of freedom are} R-groups that have rotational torsion symmetry.

Text

We are very grateful for the reviewer’s careful reading of our manuscript.

Page 2, first paragraph: “if the energy difference” should be “free energy difference”

Done.

Page 3, third paragraph: “those that measures” should be “measure”

Done.

Page 5, second paragraph: “all others uses 12” should be “use”

Done.

Page 6, last paragraph: “Pauslon et al” should be “Paulsen et al”

Done.

Page 7, last paragraph: “that has an RMSEs” could be “that has an RMSE”

Done.

Page 8, first paragraph: “based experimental error” could be “based on experimental error”

Done.

Page 8, last paragraph: “a t-distributions” could be “a t-distribution”

Done.

Page 9, first paragraph: “imply a pairwise RMSEs” could be “imply pairwise RMSEs”

Done.

Page 9, first paragraph: “top-right scatter plot” should it say “top-left”?

Yes, thank you. Done.

Page 10, first paragraph: “lower than the pairwise RMSE between two functional assays” should it say “binding assays”?

Yes, thank you. Done.

Page 10, table caption: “differences we have obtained is not” should be “are not”

Done.

Page 10, last paragraph: “as the data as prior publications” probably “as the data” is not needed here

Yes, thank you. Done.

Page 12, first paragraph: “absolute binding free protocol” should be “absolute binding free energy protocol”

Done.

Reviewers' comments:

Reviewer #1 (Remarks to the Author):

The authors appear to have dealt with most of my earlier comments; however, the cited benchmarking work of Hahn et al. provides what is essentially a "community consensus" as to what should be done in benchmarking, or at least an attempt towards one. It would be very nice to see the authors of this work either (a) engage with Hahn and co-authors (who do represent a broad swath of the community!) to see if they can arrange to modify that consensus, or (b) conform to the consensus. Instead, they have done neither and their response seems to be "Nevermind, we don't think the consensus applies here". Given that the authors of the Hahn et al. article cover a broad swath of industry and computational chemistry (far broader than the present authors list or even the reviewers) it would be nice to see the authors at least engage with them.

That said, my objection in this regard does not seem like it should warrant holding this paper back.

Reviewer #2 (Remarks to the Author):

The authors have worked thoroughly to address all the concerns raised. I fully recommend the article for publication in its current form.

Reviewer #3 (Remarks to the Author):

Summary and recommendation

I am glad to see that the authors took my comments into consideration. Overall, I am satisfied with the changes made and have only one concern regarding the first major comment from the previous round.

In particular, I am not convinced by the arguments against providing any insight into the variation among the computational results. In principle, the authors make a claim that computational procedures that differ from theirs will inherently be less accurate and should not be taken into account. However, at the same time the authors compile values from disparate experiments with differing assay conditions, sample preparations, measurement techniques etc to report on the experimental variability. Following the current argument, a proponent of a particular experimental approach could request to use his/her approach only, as it might be the most accurate one. This is of course just a hypothetical example, but it illustrates that the experimental and computational data is not treated equivalently in the current work. In my view, such a different treatment of computational and experimental results should be avoided. If the authors insist on including computational results obtained with their software only, it should still be possible to collect from literature a number of repeated estimations for the same systems performed by independent groups.

All in all, I think that after addressing this concern the manuscript can be accepted for publication in Communications Chemistry.

2nd Review comments

Reviewer #1 (Remarks to the Author):

The authors appear to have dealt with most of my earlier comments; however, the cited benchmarking work of Hahn et al. provides what is essentially a "community consensus" as to what should be done in benchmarking, or at least an attempt towards one. It would be very nice to see the authors of this work either (a) engage with Hahn and co-authors (who do represent a broad swath of the community!) to see if they can arrange to modify that consensus, or (b) conform to the consensus. Instead, they have done neither and their response seems to be "Nevermind, we don't think the consensus applies here". Given that the authors of the Hahn et al. article cover a broad swath of industry and computational chemistry (far broader than the present authors list or even the reviewers) it would be nice to see the authors at least engage with them.

That said, my objection in this regard does not seem like it should warrant holding this paper back.

We thank the reviewer for their comments. We intend to fully engage with the wider community, including the authors Hahn et al., and view the publication of our work in a peer reviewed journal as an excellent starting point to discussion and debate. We look forward to this discussion and reaching a consensus in the future.

Reviewer #2 (Remarks to the Author):

The authors have worked thoroughly to address all the concerns raised. I fully recommend the article for publication in its current form.

We thank the reviewers for their comments and are happy to know we addressed all of their concerns.

Reviewer #3

I am glad to see that the authors took my comments into consideration. Overall, I am satisfied with the changes made and have only one concern regarding the first major comment from the previous round. In particular, I am not convinced by the arguments against providing any insight into the variation among the computational results. In principle, the authors make a claim that computational procedures that differ from theirs will inherently be less accurate and should not be taken into account. However, at the same time the authors compile values from disparate experiments with differing assay conditions, sample preparations, measurement techniques etc to report on the experimental variability. Following the current argument, a proponent of a particular experimental approach could request to use his/her approach only, as it might be the most accurate one. This is of course just a hypothetical example, but it illustrates that the

experimental and computational data is not treated equivalently in the current work. In my view, such a different treatment of computational and experimental results should be avoided. If the authors insist on including computational results obtained with their software only, it should still be possible to collect from literature a number of repeated estimations for the same systems performed by independent groups.

All in all, I think that after addressing this concern the manuscript can be accepted for publication in Communications Chemistry.

We are grateful to the reviewer for the comments and suggestions and are glad that the reviewer is satisfied with the changes we have made. With regards to the first major comment about the variance in computational results, we thank the reviewer for clarifying this further and have taken the opportunity to further improve the manuscript with the additional analysis of the variance of FEP+ results from different approaches. We hope that the reviewer agrees with our discussion below as well as the additions we have made to the manuscript.

First, the purpose of comparing the experimental results from different assays is to provide an estimate of the upper limit of computational accuracy. In the benchmark, the accuracy of free energy perturbation (FEP) predictions is calculated using the measurements from one assay; the type of assay (e.g. functional or binding) varies from system to system and, in the majority of cases, the quality of the assay is completely unknown to us. Indeed, there is no doubt that there is a broad range of assay quality used to compute FEP accuracy in our benchmark. The variations of experimental results from all different assays imply an upper limit of the accuracy for free energy calculations. Ideally, our experimental reproducibility survey is big enough to have an equivalently broad array of assay quality so that the experimental error estimates (e.g. RMSE, R^2) are directly comparable to the maximal FEP accuracy. Strictly speaking, this estimate of the maximal accuracy of FEP applies to cases when the FEP predictions are compared against measurements from an equivalent distribution of assay quality as our experimental reproducibility survey. This point may not have been clearly conveyed in our manuscript, so we have added the following sentences to the conclusion (page 12) of the main text.

It should also be noted that given the heterogeneity of the assay quality used both in the experimental survey and FEP benchmark, our estimates of the maximal and current accuracy primarily apply to this same regime of assay quality.

Please note that we did not seek out lower quality assays in either the FEP benchmark or experimental survey, but these are bound to be present given the size of the studies.

Second, we agree that there may be a general interest in estimating the variance between computational predictions and we made the best effort to provide such an estimate. Since FEP+ is a commercial product offered by Schrödinger, a comparison of FEP+ to other methods may be perceived as unfair and generate potential conflict of interest. To avoid such concerns,

we will only use FEP+ as performed by Schrödinger in our estimate for the variance of computational results, and sincerely hope the reviewer understands this sensitive point.

For our estimate of the variance arising from different computational methods, we used the data set referred to as the “FEP+ R-group set” in the manuscript (that is also colloquially known as the “JACS set”). This data set has been a standard benchmark for FEP+ since 2015, and comprises 199 ligands across 8 targets. While there have been minor adjustments to the structures since 2015, it has remained a very stable data set that has been widely used in the community. We use the results from the original 2015 publication and 3 subsequent force field development papers (up to 2021) for the comparison (note that significant improvements to JNK1, MCL1 and thrombin data sets including input models to FEP+ were made in the current study thus the results from this paper are not used in the comparison). The sampling methodology and force field underwent significant improvements between 2015 and 2021, such as enhanced water sampling and improved charged interactions. The publications from 2015 to 2021 also used different amounts of sampling time. The predictions from 2015 up to 2021 reflect results from “modern” methods. We believe that the results from 2015 up to 2021 would be a fair reflection of the variance of “modern” FEP computational approaches, given the changes that FEP+ went through during that time.

For this study, we manually extracted the raw predictions from the original publications and reanalysed them using exactly the same metrics and statistics that have been used in the rest of our manuscript. (The metrics used in the prior publications were not wholly consistent with the metrics used in this manuscript.) The table of results are shown in Table S33 of the supporting information. The spread of the metrics across the 4 sets of predictions is our estimate of the variance of a FEP method on this data set. For instance, as the weighted pairwise RMSE ranges from 1.09 kcal/mol to 1.34 kcal/mol, our estimate of the variance of this metric that comes from different computational methods (on this data set) is approximately 0.25 kcal/mol.

We also extracted the data from the 2021 OPLS4 publication and re-analysed the results using the same metrics as used in this manuscript. The data set used to validate OPLS4 contains 512 protein-ligand pairs (compared to the JACS set 199) and is far more heterogeneous in terms of its target and perturbation type coverage. We observe a larger difference in accuracy between OPS3e and OPLS4 compared to the difference in accuracy in the JACS set (Table S34 in SI). This indicates that larger, more diverse data sets can increase the apparent variance in the predictions from different computational methods.

This new analysis can be found in section 3 of the SI (page 33-34) and section 4.2, page 8 of the main text. The new paragraph in the main text is as follows:

FEP accuracy varies depending on the method used for the calculation. To quantify this variability and the degree to which this is affected by the size and heterogeneity of the data set, we re-analysed the predictions from FEP+ predictions from 2015 up to 2021, which encompasses sampling and force field improvements. The results, in section 3 of

the SI, indicate that larger, more diverse data sets are more able to discriminate between the accuracy of different approaches.

REVIEWERS' COMMENTS:

Reviewer #3 (Remarks to the Author):

I am satisfied with the reply and revision and recommend the manuscript for publication in Communications Chemistry.